# Resistance to different anthracycline chemotherapeutics elicits distinct and actionable primary metabolic dependencies in breast cancer

Shawn McGuirk[1,2], Yannick Audet-Delage[3,4], Matthew G Annis[2,5], Yibo Xue[1,2], Mathieu Vernier[2], Kaiqiong Zhao[6,7], Catherine St-Louis[3,4], Lucía Minarrieta[3,4], David A Patten[3,4], Geneviève Morin[1,2], Celia MT Greenwood[6,7,8,9], Vincent Giguère[1,2], Sidong Huang[1,2], Peter M Siegel[2,5], Julie St-Pierre[1,2,3,4]*

[1]Department of Biochemistry, Faculty of Medicine, McGill University, Montreal, Canada; [2]Goodman Cancer Research Centre, McGill University, Montreal, Canada; [3]Department of Biochemistry, Microbiology, and Immunology, University of Ottawa, Ottawa, Canada; [4]Ottawa Institute of Systems Biology, Ottawa, Canada; [5]Department of Medicine, Faculty of Medicine, McGill University, Montreal, Canada; [6]Department of Epidemiology, Biostatistics and Occupational Health, McGill University, Montreal, Canada; [7]Lady Davis Institute, Jewish General Hospital, Montreal, Canada; [8]Department of Human Genetics, McGill University, Montreal, Canada; [9]Gerald Bronfman Department of Oncology, Montreal, Canada

*For correspondence:
julie.st-pierre@uottawa.ca

Competing interests: The authors declare that no competing interests exist.

**Abstract** Chemotherapy resistance is a critical barrier in cancer treatment. Metabolic adaptations have been shown to fuel therapy resistance; however, little is known regarding the generality of these changes and whether specific therapies elicit unique metabolic alterations. Using a combination of metabolomics, transcriptomics, and functional genomics, we show that two anthracyclines, doxorubicin and epirubicin, elicit distinct primary metabolic vulnerabilities in human breast cancer cells. Doxorubicin-resistant cells rely on glutamine to drive oxidative phosphorylation and *de novo* glutathione synthesis, while epirubicin-resistant cells display markedly increased bioenergetic capacity and mitochondrial ATP production. The dependence on these distinct metabolic adaptations is revealed by the increased sensitivity of doxorubicin-resistant cells and tumor xenografts to buthionine sulfoximine (BSO), a drug that interferes with glutathione synthesis, compared with epirubicin-resistant counterparts that are more sensitive to the biguanide phenformin. Overall, our work reveals that metabolic adaptations can vary with therapeutics and that these metabolic dependencies can be exploited as a targeted approach to treat chemotherapy-resistant breast cancer.

## Introduction

Therapeutic resistance is a central problem in the clinical treatment of cancer. The incidence of breast cancer has risen to over one million new cases per year worldwide, where 20–30% of cases are diagnosed at an advanced or metastatic stage and another 30% recur or develop metastases (*Siegel et al., 2018*; *Murray et al., 2012*). While both adjuvant and neoadjuvant therapies have proven effective to improve patient outcomes, not all patients respond to the same therapeutics. Furthermore, drug resistance can manifest within months of treatment and is believed to cause treatment failure in over 90% of metastatic cancers (*Garrett and Arteaga, 2011*; *Longley and Johnston,*

*2005*). Consequently, due to intrinsic or acquired resistance, breast cancer patients often suffer disease progression despite drug treatment (*Murphy and Seidman, 2009*; *Moreno-Aspitia and Perez, 2009*).

In the absence of targeted therapies, chemotherapy is a standard-of-care treatment for many aggressive breast cancers (*Lebert et al., 2018*). While this is efficient at killing fast-growing cells, it can also select for resistant cells or elicit adaptations that confer resistance in surviving populations. These may include genetic modulation of mechanisms that decrease intracellular drug concentration, like drug export through the ATP-binding cassette (ABC) transporter family (*Hembruff et al., 2008*) or lysosomal clearance (*Guo et al., 2016*), as well as adaptations that minimize or overcome therapy-associated insults like DNA damage or reactive oxygen species (ROS) (*Morandi and Indraccolo, 2017*). Importantly, therapeutic agents elicit diverse resistance-conferring adaptations both across tumor subtypes and within tumors due to genetic and metabolic heterogeneity (*Caro et al., 2012*; *Viale et al., 2015*).

Several recent reviews have emphasized the importance of metabolic adaptations in driving or supporting drug resistance (*Morandi and Indraccolo, 2017*; *Viale et al., 2015*; *Wolf, 2014*; *Bosc et al., 2017*; *Ashton et al., 2018*). Although glycolysis is likely to remain favored in resistant cancers undergoing hypoxia or with defective mitochondria (*Xu et al., 2005*; *Zhou et al., 2012*), increased reliance on mitochondrial energy metabolism and oxidative phosphorylation has been identified as a distinctive characteristic of drug resistance (*Bosc et al., 2017*) being central to therapeutic resistance in ovarian (*Matassa et al., 2016*), pancreatic (*Viale et al., 2014*), colon (*Vellinga et al., 2015*), prostate (*Ippolito et al., 2016*), melanoma (*Vazquez et al., 2013*), and breast (*Lee et al., 2017*) cancers, as well as large B cell lymphoma (*Caro et al., 2012*) and acute (*Farge et al., 2017*) or chronic (*Kuntz et al., 2017*) myeloid leukemia.

Despite these advances in our understanding of the metabolic status of treatment-resistant cancers, little is known about the impact of different therapeutic drugs on the metabolic status of a given cancer. Addressing this knowledge gap is important, as numerous monotherapy and combination therapy regimens are often available to treat each patient. Here, we show that breast cancer cells resistant to either doxorubicin or epirubicin, two anthracycline drugs that are used interchangeably for breast cancer treatment (*Mao et al., 2019*), rely on distinct primary metabolic processes, and that exploiting these dependencies may impair the growth of treatment resistant breast cancers.

## Results

### Doxorubicin-resistant and epirubicin-resistant breast cancer cells display distinct global metabolic alterations

As experimental models, we used well-established and published models of breast cancer therapeutic resistance (*Hembruff et al., 2008*; *Guo et al., 2016*; *Veitch et al., 2009*; *Heibein et al., 2012*). Briefly, these models were generated from human MCF-7 breast cancer cells, adapted to increasing concentrations of either doxorubicin or epirubicin to a maximum tolerated dose of 98.1 nM (DoxR cells) or 852 nM (EpiR cells), respectively, in a stepwise manner and over several months (*Hembruff et al., 2008*; *Figure 1a*). Parental control cells (Control cells) were maintained in DMSO throughout the extensive selection process (*Hembruff et al., 2008*; *Figure 1a*). At these maximally tolerated doses, it has been shown that resistance is not simply linked to drug exclusion by the cells. Indeed, augmenting intracellular drug levels by inhibiting ABC transporter activity in DoxR or EpiR has little effect on cell survival, highlighting the importance of adaptation mechanisms separate from that of the ABC transporters and independent of drug concentration (*Hembruff et al., 2008*).

DoxR and EpiR cells maintained in culture with a stable dose of their respective drug grew slower than Control cells without treatment (*Figure 1b*), and acute exposure of Control cells to 98.1 nM of doxorubicin had a cytostatic effect, while treatment with 852 nM epirubicin was cytotoxic (*Figure 1b,c*). Finally, we verified that DoxR and EpiR cells are stably resistant, retaining their level of resistance even after a 7-week drug holiday (*Figure 1—figure supplement 1a*).

In line with the chemical similarity and mechanism of action of both anthracycline drugs—nucleic acid intercalation, topoisomerase II inhibition leading to double-strand DNA breaks and apoptosis, and production of ROS (*McGowan et al., 2017*)—there was a considerable overlap in the signature

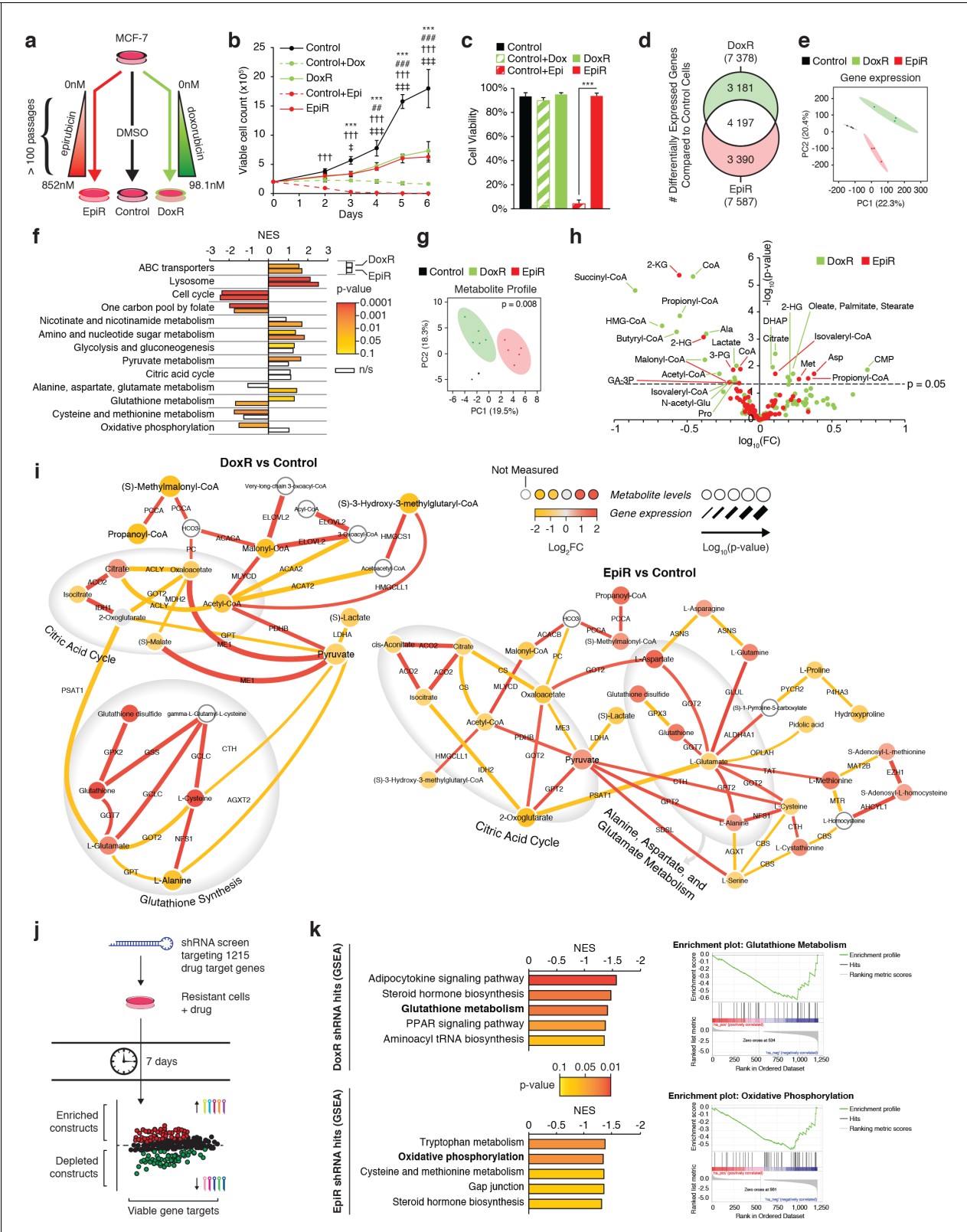

**Figure 1.** Transcriptomic and metabolomic analyses uncover distinct metabolic alterations in doxorubicin and epirubicin resistant breast cancer cells, compared to anthracycline-sensitive Control cells. (a) Model detailing stepwise generation of Control, DoxR, and EpiR breast cancer cells, as previously described (*Hembruff et al., 2008*). (b) Viable cell number of anthracycline-resistant cells in the presence of drugs and anthracycline-treated Control cells compared to untreated Control (DMSO). N = 4, ***p<0.001 Control vs Control +Dox, ##p<0.01 ###p<0.001 DoxR vs Control +Dox, †††p<0.001

*Figure 1 continued on next page*

Figure 1 continued

Control vs Control +Epi, ‡p<0.05 ‡‡‡p<0.001 EpiR vs Control +Epi (two-way ANOVA, Tukey's multiple comparison test). (c) Viability of anthracycline-resistant cells and treated or untreated (DMSO) Control cells after 7 days growth. N = 4, ***p<0.001 (paired Student's t-test). (d) Venn diagram of differentially expressed genes in DoxR or EpiR cells compared to Control cells. Legend: green (differentially expressed in DoxR only), red (differentially expressed in EpiR only), and white (differentially expressed in both). N = 3. (e) Principal component analysis of global gene expression profiling between Control, DoxR, and EpiR cells. N = 3. (f) Gene set enrichment analysis (GSEA) of DoxR and EpiR cells compared to Control cells (KEGG database, N = 3). Data are shown as normalized enrichment score (NES) for DoxR and EpiR, where color designates the p-value associated with each enrichment (from yellow at p=0.1 to red at p=0.0001) and where white bars designate non-significant enrichments (n/s, p>0.1). (g) Partial least squares discriminant analysis of metabolite profile data between Control, DoxR, and EpiR cells. N = 6, p=0.008 (1000 permutations). (h) Volcano plot of metabolite profile of DoxR and EpiR cells compared to Control cells (N = 6). Significant features (p<0.05, paired Student's t-test) highlighted. (i) Integrated metabolic network analysis of DoxR (left) and EpiR (right) cells compared to Control cells. Metabolites are represented by nodes, with p-value represented by node size. Enzymes are represented by edges, with p-value of gene expression represented by edge thickness. Direction and magnitude of fold changes in gene expression and metabolite abundance are represented on a yellow (depleted in resistant) to red (enriched in resistant) color scale. Major pathways are highlighted in shaded areas. (j) Model detailing shRNA screen targeting 1215 drug target genes. Changes in shRNA barcode abundance were measured after 7 days, and viable gene targets were ranked by p-value and fold change of target shRNA abundance. Depleted shRNA were considered cytotoxic or cytostatic to resistant cells in the presence of drug, while enriched barcodes were considered to promote cell proliferation or survival. (k) (left) Top five metabolic pathways identified by Gene Set Enrichment Analysis (KEGG database) of ranked shRNA gene targets depleted in DoxR (top) and EpiR (bottom) cells. (right) GSEA plots detailing enrichment of key metabolic pathways identified in DoxR cells (Glutathione Metabolism, top) or EpiR (Oxidative Phosphorylation, bottom). All data presented as averages ± S.E.M. (b-c).

The online version of this article includes the following figure supplement(s) for figure 1:

**Figure supplement 1.** Common and distinct pathways modulated in DoxR and EpiR cells compared to Control cells.

of differentially expressed genes between DoxR and EpiR cells when compared to Control cells (*Figure 1d*, 56% overlap). These gene expression signatures also resemble that of human breast cancer tumors after epirubicin-based therapy where biopsies were taken prior to and after treatment, consisting of four cycles of epirubicin and cyclophosphamide, followed by four cycles of docetaxel (*Figure 1—figure supplement 1b,c*; GSE43816) (*Gruosso et al., 2016*). While the surviving tumor fractions in this dataset cannot be assumed to be entirely treatment-resistant, they represent a close clinical approximate of the adaptations that may occur after any anthracycline-based therapy, or as a result of selected advantages that may promote resistance. Gene expression profiles of both DoxR and EpiR cells significantly overlapped with that of post-treatment biopsies, when compared to the overlap expected from random shuffling (*Figure 1—figure supplement 1b,c*). The overlap in the gene expression profiles between DoxR and EpiR cells as well as that between the respective expression profile of DoxR and EpiR cells with breast cancer patients post-treatment biopsies highlight common mechanisms of adaptation to anthracyclines as well as the clinical relevance of the DoxR and EpiR breast cancer cell models.

Despite these similarities, we noted that 44% of differentially expressed genes in DoxR and EpiR cells were distinct (*Figure 1d*). A principal component analysis of their global gene expression profiles accordingly produced three discrete groups, with DoxR and EpiR diverging from each other in the second principal component (*Figure 1e*). Gene Set Enrichment Analysis (GSEA) further revealed that, although both DoxR and EpiR display enrichment in drug clearance pathways (ABC transporters, lysosome) and depletion of pathways supporting proliferation (cell cycle, one carbon pool by folate) compared to Control cells, several metabolic pathways are specifically enriched in DoxR (pyruvate metabolism, glutathione metabolism) or EpiR (nicotinate and nicotinamide metabolism, alanine, aspartate, and glutamate metabolism) (*Figure 1f*, *Figure 1—figure supplement 1d*).

Analysis of the global metabolite profiles of Control, EpiR, and DoxR also indicated significant differences between all three models (*Figure 1g,h*). To visualize the distinct metabolic adaptations that occur in resistance to either doxorubicin or epirubicin, we performed an integrated transcriptional and metabolic network analysis (*Sergushichev et al., 2016*) of DoxR and EpiR cells compared to parental Control cells (*Figure 1i*). DoxR cells displayed increased expression of glutathione pathway genes as well as elevated levels of key glutathione metabolites (glutamate, cysteine, glutathione disulfide), indicating a likely role of this pathway in overcoming doxorubicin-induced oxidative stress (*Figure 1i*; *Pilco-Ferreto and Calaf, 2016*). EpiR cells showed alterations in pathways linked to pyruvate and glutamate metabolism, in particular through elevated transamination reactions linking glutamate, alanine, and aspartate (*Figure 1i*).

In parallel to these transcriptomics and metabolomics analyses, we performed pooled shRNA screens focused on 1215 druggable genes to identify primary vulnerabilities of DoxR and EpiR cells (*Figure 1j*). In this screen, the enrichment or depletion of shRNA barcodes in each cell system is measured over 7 days (post-integration of shRNAs); constructs whose barcodes are depleted over this span indicate gene targets whose knockdown impairs growth and/or cell viability. Analyzing depleted constructs via GSEA determined that DoxR cells were particularly sensitive to knockdown of glutathione metabolism genes (*GSR* and *GPX* family genes), while EpiR cells were vulnerable to suppression of genes involved in oxidative phosphorylation (*NDUF* and *SDH* family genes) and methionine metabolism (*MAT1A, MAT2A, MAT2B, BHMT, DNMT1*; *Figure 1k* and *Figure 1—figure supplement 1e*). In agreement with previous work (*Veitch et al., 2009*; *Heibein et al., 2012*), both DoxR and EpiR cells were sensitive to knockdown of aldo-keto reductase family genes, which are represented in the steroid hormone biosynthesis pathway (*Figure 1k*). Overall, results from the shRNA screens are consistent with our integrated transcriptional and metabolic network analysis (*Figure 1i*) and highlight distinct metabolic vulnerabilities supporting epirubicin and doxorubicin resistance.

## Doxorubicin-resistant breast cancer cells display altered glucose and glutamine metabolism

To gain greater understanding of the reliance of anthracycline-resistant breast cancer cells on the pathways identified in the integrated analyses above, we confirmed gene expression profiles by RT-qPCR (*Figure 2a*) and performed stable isotope tracer analyses of [U-$^{13}$C]-glucose (*Figure 2b–d*, *Figure 2—figure supplement 1a,b*) and [U-$^{13}$C]-glutamine (*Figure 2e,f*, *Figure 2—figure supplement 2a,b*). The full kinetics of all stable isotope tracing experiments are shown in *Figure 2—figure supplements 1* and *2*, in accordance with the standard practice in the field (*Buescher et al., 2015*).

DoxR cells exhibited significantly increased expression of anaplerotic metabolism genes (*PC, ME1, ME2*), glutamine metabolism genes (*SLC1A5, GLS, GLUL*), and, markedly, glutathione metabolism genes (*GCLC, GCLM, GSS, GSR*) compared to Control cells (*Figure 2a*). Accordingly, kinetic tracing of glucose carbons showed that while Control and EpiR cells replenish their pools of citric acid cycle intermediates (citrate and malate m + 2) principally through pyruvate dehydrogenase (PDH), DoxR cells significantly favor anaplerotic pyruvate metabolism (citrate, malate, and fumarate m + 3) through pyruvate carboxylase (PC) and/or malic enzymes (ME1/2; *Figure 2d* and *Figure 2—figure supplement 1a,b*). Interestingly, glutamate, alanine, and serine synthesis from glucose was decreased in DoxR cells compared to Control cells (*Figure 2d* and *Figure 2—figure supplement 1a,b*). Kinetic tracing further showed that glutamine metabolism was enriched in DoxR cells compared to Control and EpiR cells, evidenced by increased labeling to glutamate, α-ketoglutarate, and citrate (*Figure 2e,f* and *Figure 2—figure supplement 2a,b*). Reductive carboxylation of glutamine was particularly increased in DoxR cells compared to Control and EpiR cells, as indicated by a significant increase in m + 5 labeling to citrate (Figre 2e,f and *Figure 2—figure supplement 2a,b*). More strikingly, DoxR cells largely favored the use of glutamine carbons for *de novo* production of glutathione, evidenced by a fourfold enrichment of labeling to GSH and GSSG compared to both Control and EpiR cells (GSH m + 5, GSSG m + 5,10; *Figure 2f* and *Figure 2—figure supplement 2a,b*). This may also be driven in part by exchange of glutamate for cystine through the glutamate/cystine antiporter system (*Habib et al., 2015*) as DoxR cells were found to export significantly higher levels of glutamate than both Control and EpiR cells (*Figure 2—figure supplement 3a*).

In agreement with these stable isotope tracing results, DoxR cells displayed a significantly higher total intracellular glutathione concentration than EpiR cells, and both had higher values than Control cells (*Figure 2g*). Both resistant lines had a significantly higher GSH:GSSG ratio than Control cells, with DoxR cells displaying an even greater enrichment of reduced glutathione compared to EpiR cells (*Figure 2h*). The elevated glutathione metabolism in DoxR cells is further supported by their decreased NADH:NAD and NADPH:NADP ratios (*Figure 2i*) compared to Control cells, as the reduced equivalent NADPH is required for the reduction of GSSG to GSH through glutathione reductase (GSR), whose expression was increased in DoxR but not EpiR cells compared to Control cells (*Figure 2a*). NADPH levels may also be depleted through the reductive carboxylation of glutamine, as this pathway relies on the NADPH-dependent isocitrate dehydrogenases (IDH1/2); the expression of IDH1 is significantly increased in DoxR cells compared to both EpiR and Control cells, in line with their increased engagement of this pathway (*Figure 2a,f*). Conversely, EpiR cells

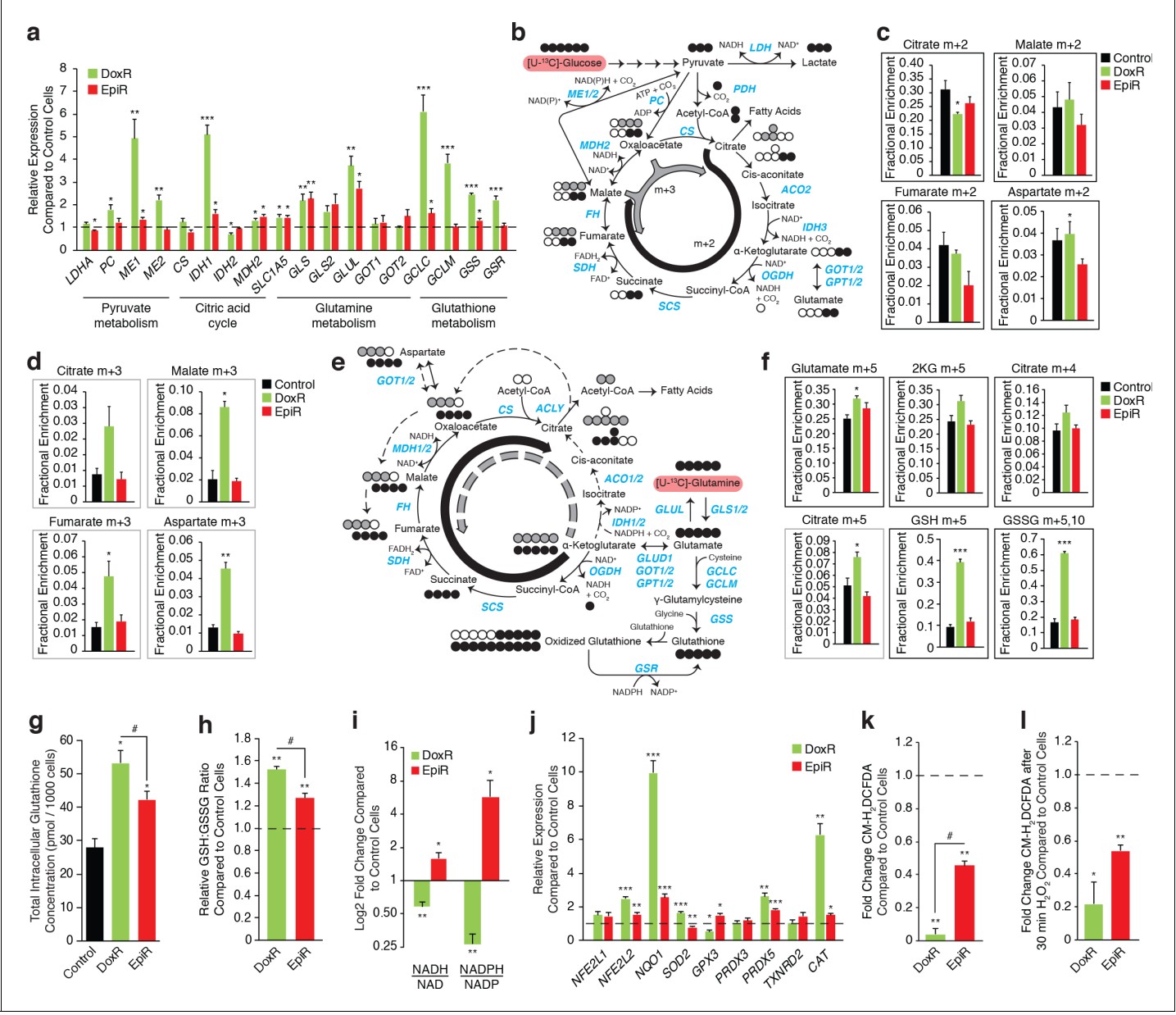

**Figure 2.** Doxorubicin-resistant breast cancer cells fuel anaplerotic metabolism by altering glucose and glutamine metabolism. (a) Relative expression of pyruvate metabolism, citric acid cycle, and glutamine and glutathione metabolism genes in DoxR and EpiR compared to Control cells. N = 3–6, *p<0.05 **p<0.01 ***p<0.001 resistant versus Control cells (paired Student's t-test). (b) Stable isotope tracing diagram for [U-$^{13}$C]-Glucose through glycolysis and into the citric acid cycle via pyruvate dehydrogenase (PDH, black) and adjacent pathways (gray). (c) [U-$^{13}$C]-Glucose tracing into the citric acid cycle via PDH (citrate, malate, fumarate, and aspartate m + 2, 30 min tracer) in Control, DoxR, and EpiR cells expressed as fractional enrichment. N = 4, *p<0.05 resistant versus Control cells (paired Student's t-test). (d) [U-$^{13}$C]-Glucose tracing to glutamate (m + 2) via PDH and into the citric acid cycle via PC or ME1/2 activity (citrate, malate, fumarate, and aspartate m + 3, 30 min tracer) in Control, DoxR, and EpiR cells expressed as fractional enrichment. N = 4, *p<0.05 **p<0.01 resistant versus Control cells (paired Student's t-test). (e) Stable isotope tracing diagram for [U-$^{13}$C]-Glutamine into the citric acid cycle, glutathione synthesis, and adjacent pathways. Reductive carboxylation pathway shown in gray. (f) [U-$^{13}$C]-Glutamine tracing to glutamate and into the citric cycle (2KG m + 5, citrate m + 4 and m + 5, 60 min tracer) or through to glutathione (GSH m + 3,5 and GSSG m + 3,5,6,8,10, 4 hr tracer) in Control, DoxR, and EpiR cells expressed as fractional enrichment. N = 4, *p<0.05 ***p<0.001 resistant versus Control cells (paired Student's t-test). (g) Total intracellular glutathione concentration in Control, DoxR and EpiR cells. N = 4, *p<0.05 resistant vs Control cells, #p<0.05 DoxR vs EpiR cells (paired Student's t-test). (h) Fold change in GSH:GSSG ratio of DoxR and EpiR cells compared to Control cells. N = 4, **p<0.01 resistant vs Control, #p<0.05 DoxR vs EpiR cells (paired Student's t-test). (i) Fold change in NADH/NAD and NADPH/NAD ratio of DoxR and EpiR cells compared to Control cells. N = 6, *p<0.05 **p<0.01 resistant versus Control cells (paired Student's t-test). Data are shown on a log$_2$ scale. (j) Relative expression of oxidative response genes in DoxR and EpiR cells compared to Control cells. N = 4–7, *p<0.05 **p<0.01 ***p<0.001 resistant vs Control cells (paired Student's t-test). (k) Fold change of ROS signal in DoxR and EpiR cells compared to Control cells, detected by CM-H$_2$DCFDA.

*Figure 2 continued on next page*

*Figure 2 continued*

N = 3, **p<0.01 resistant vs Control cells, #p<0.05 DoxR vs EpiR cells (paired Student's t-test). (I) Fold change of ROS signal in DoxR and EpiR compared to Control cells, after 30-min treatment with 0.03% $H_2O_2$. N = 3, *p<0.05 **p<0.01 resistant vs Control cells (paired Student's t-test). All data presented as averages ± S.E.M.

The online version of this article includes the following figure supplement(s) for figure 2:

**Figure supplement 1.** [U-$^{13}$C]-glucose tracing of Control, DoxR, and EpiR breast cancer cells.

**Figure supplement 2.** [U-$^{13}$C]-glutamine tracing of Control, DoxR, and EpiR breast cancer cells.

**Figure supplement 3.** Media metabolite composition of Control, DoxR, and EpiR breast cancer cells.

displayed elevated NADH:NAD and NADPH:NADP ratios compared to Control cells (*Figure 2i*). EpiR cells also did not display any significant changes in either anaplerotic pyruvate metabolism (*Figure 2d* and *Figure 2—figure supplement 1a,b*), reductive carboxylation of glutamine, or *de novo* glutathione synthesis from glutamine (*Figure 2f* and *Figure 2—figure supplement 2a,b*) when compared to Control cells, even though they showed increased expression of glutamine metabolism genes (*Figure 2a*).

Given that both DoxR and EpiR cells had a higher GSH:GSSG ratio compared to Control cells, we further sought to determine whether these anthracycline-resistant cells displayed markers of elevated oxidative stress response. Both resistant lines had increased expression of key antioxidant genes compared to Control cells, with DoxR cells showing a higher increase in expression of many genes (*NFE2L2*, *NQO1*, *SOD2*, *PRDX5*, *CAT*) compared with EpiR cells (*Figure 2j*). Through CM-H$_2$DCFDA experiments, we also determined that DoxR cells have decreased ROS signals compared to EpiR cells, both at baseline (*Figure 2k*) and after $H_2O_2$ treatment (*Figure 2l*), while both resistant cells had lower ROS signals than Control cells. DoxR cells likely support this greater engagement of oxidative stress response through their increased *de novo* glutathione synthesis from glutamine, in agreement with findings from the shRNA screen showing that glutathione metabolism is a specific vulnerability in this model (*Figure 1k*).

## Epirubicin-resistant breast cancer cells display increased oxidative bioenergetic capacity

Given that oxidative phosphorylation was identified as a specific vulnerability for EpiR cells in the shRNA screen (*Figure 1k*), we sought to determine the bioenergetic profile of these cells. Using the Seahorse platform, we found that basal and maximal oxygen consumption rates were significantly increased in EpiR cells, but not DoxR cells, compared to Control cells, whereas extracellular acidification rates were not significantly different amongst the three cell lines, albeit slightly lower in EpiR cells (*Figure 3a–c*). From these data, we further extrapolated the total rates of ATP production ($J_{ATP}$) as well as the fraction of ATP generated through glycolysis ($J_{ATPglyc}$) or oxidative phosphorylation ($J_{ATPox}$) using published assumptions and algorithms (*Mookerjee et al., 2017*). Aligning with their increased respiration, a greater proportion of ATP produced in EpiR cells was linked to oxidative phosphorylation — 69%, compared to 62% and 60% for Control and DoxR cells respectively (*Figure 3d,e*). However, the total ATP production rate in EpiR cells was not significantly different than Control cells, as their glycolyic ATP production was proportionally lowered (*Figure 3d*). EpiR cells also acquired a significant increase in total bioenergetic capacity (25%) compared to both Control and DoxR cells (*Figure 3f,g*), largely driven by increased oxidative capacity (*Figure 3f*) and, consequently, EpiR cells have a greater reserve capacity for generation of ATP through oxidative phosphorylation (*Figure 3h*). This increased capacity is further supported by elevated mitochondrial volume in EpiR cells compared to Control or DoxR cells (*Figure 3i*).

Given the importance of peroxisome proliferator-activated receptor gamma coactivator 1-alpha (PGC-1α) in mitochondrial biogenesis (*Wu et al., 1999*), OXPHOS (*Mootha et al., 2003*), glutamine metabolism (*McGuirk et al., 2013*), and glutathione synthesis (*Guo et al., 2018*), we hypothesized that it may play a role in the metabolic adaptations that support doxorubicin and/or epirubicin resistance. Indeed, *PPARGC1A* expression at the mRNA (*Figure 3j*) and protein (*Figure 3k*) levels was markedly increased in EpiR compared to Control cells, while there was a modest and non-significant increase in DoxR compared to Control cells. mRNA expression of ERRα (*ESRRA*), the central transcription partner of PGC-1α, was also increased in DoxR and EpiR cells compared to Control cells

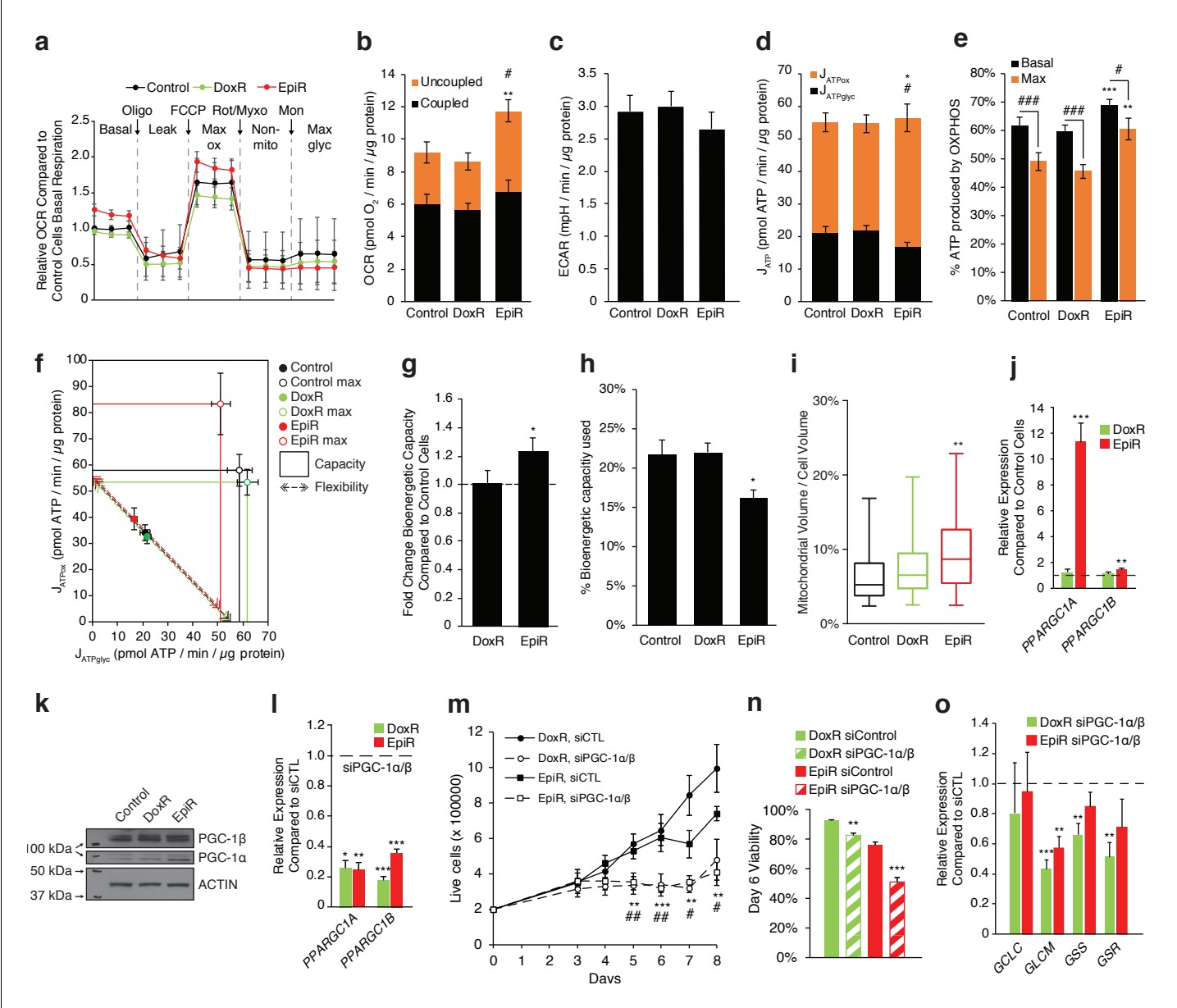

**Figure 3.** PGC-1α is overexpressed and elevates OXPHOS capacity in epirubicin-resistant cells but is essential for sustaining growth and survival in both doxorubicin and epirubicin resistant breast cancer cells. (**a**) Analysis of relative basal, leak (oligomycin), maximum (FCCP), and non-mitochondrial (rotenone and myxothiazol) respiration, as well as after addition of monensin, in Control, DoxR, and EpiR cells. All values normalized to basal respiration in Control cells. N = 9. (**b**) Quantification of coupled, uncoupled, and total Oxygen Consumption Rate (OCR) in Control, DoxR, and EpiR cells. N = 9, **p<0.01 uncoupled #p<0.05 total, resistant vs Control cells (paired Student's t-test). (**c**) Quantification of basal Extracellular Acidification Rate (ECAR) in Control, DoxR, and EpiR cells. N = 9. (**d**) Quantification of ATP production rate ($J_{ATP}$) from oxidative phosphorylation ($J_{ATPox}$) or glycolysis ($J_{ATPglyc}$) in Control, DoxR, and EpiR cells. N = 9, *p<0.05 $J_{ATPox}$ #p<0.05 $J_{ATPglyc}$, resistant vs Control cells (paired Student's t-test). (**e**) Proportion of ATP produced by OXPHOS in Control, DoxR, and EpiR cells under basal conditions and at peak bioenergetic capacity (Max). N = 9, **p<0.01 ***p<0.001 resistant vs Control cells, #p<0.05 ##p<0.01 ###p<0.001 Max vs Basal (paired Student's t-test). (**f**) Bioenergetic space plots of Control, DoxR, and EpiR cells. Solid points represent basal $J_{ATPglyc}$ and $J_{ATPox}$ values, hollow points represent theoretical maximums for $J_{ATPox}$ (FCCP) and $J_{ATPglyc}$ (rotenone, myxothiazol, monensin). Dotted line arrows' length represents flexibility of ATP production within maximum boundaries (solid lines). Area under maximum boundaries represents the bioenergetic capacity. N = 9. (**g**) Fold change in bioenergetic capacity of DoxR and EpiR cells compared to Control cells. N = 9, *p<0.05 resistant vs Control cells (paired Student's t-test). (**h**) Fraction of bioenergetic capacity used under basal conditions in Control, DoxR, and EpiR cells. N = 9, *p<0.05 resistant vs Control cells (paired Student's t-test). (**i**) Quantification of mitochondrial volume as a percentage of total cytoplasmic volume, in Control, DoxR, and EpiR cells. N = 38, **p<0.01 resistant vs Control cells (Kruskal-Wallis test and Dunn's multiple comparisons test). Data presented as a box plot. (**j**) Relative expression of *PPARGC1A* and *PPARGC1B* mRNA in DoxR and EpiR cells compared to Control cells. N = 7, **p<0.01 ***p<0.001 resistant vs Control cells (paired Student's t-test). (**k**) Immunoblots of PGC-1β, PGC-1α, and Actin protein expression in

*Figure 3 continued on next page*

*Figure 3 continued*

Control, DoxR, and EpiR cells. (l) Relative expression of *PPARGC1A* and *PPARGC1B* mRNA in DoxR and EpiR 3 days after double siRNA knockdown of PGC-1α and PGC-1β, compared to control siRNA. N = 5, *p<0.05 **p<0.01 ***p<0.001 siPGC-1α/β vs siCTL (paired Student's t-test). (m) Growth of DoxR and EpiR in the presence of anthracyclines (dox 98.1 nM or epi 852 nM) with double siRNA knockdown of PGC-1α and PGC-1β or control siRNA. N = 5, **p<0.01 ***p<0.001 DoxR siPGC-1α/β vs DoxR siCTL, #p<0.05 ##p<0.01 EpiR siPGC-1α/β vs EpiR siCTL (paired Student's t-test). (n) Cell viability at day 6 of the growth curve shown in d. N = 5, **p<0.01 ***p<0.001 siPGC-1α/β vs siCTL (paired Student's t-test). (o) Relative expression of glutathione metabolism genes 3 days after double siRNA knockdown of PGC-1α and PGC-1β, compared to control siRNA. N = 5, **p<0.01 ***p<0.001 siPGC-1α/β vs siCTL (paired Student's t-test). All data presented as averages ± S.E.M.

The online version of this article includes the following figure supplement(s) for figure 3:

**Figure supplement 1.** PGC-1α and ERRα are enriched at the promoters of key metabolic and resistance-associated genes.

(*Figure 3—figure supplement 1a*, *Giguère, 2008*). Furthermore, both PGC-1α and ERRα were found to be enriched at the promoters of key metabolic and resistance-associated genes whose expression is modulated in DoxR or EpiR compared to Control. Specifically, they were enriched at the promoters of genes central to glutathione metabolism (*GSS*, *GSR*), oxidative response (*NQO1*, *CAT*, *NFE2L2*, *HMOX2*), drug efflux (*ABCB1*, *ABCC1*), anabolic pyruvate metabolism (*PC*), as well as *AKR1C3* and *FTH1*, two genes that were previously shown to be involved in doxorubicin resistance (*Figure 3—figure supplement 1b,c*; *Hembruff et al., 2008*; *Veitch et al., 2009*). The promoter enrichment of PGC-1α is notably higher in EpiR cells, aligning with its elevated expression level in these cells (*Figure 3—figure supplement 1b,c*).

To test the biological significance of PGC-1α in resistant cells, we opted to knock down both *PPARGC1A* and *PPARGC1B* in DoxR and EpiR cells (*Figure 3l*) to avoid potential compensation between these two transcriptional coactivators as they regulate overlapping gene expression programs (*Villena, 2015*; *Lin et al., 2005*). Knockdown of both *PPARGC1A* and *PPARGC1B* significantly abrogated the growth (*Figure 3m*) and survival (*Figure 3n*) of both DoxR and EpiR cells, compared to siCTL. Also, knockdown of PGC-1s significantly reduced the expression of key glutathione metabolism genes in DoxR (*GCLM*, *GSS*, *GSR*) and EpiR (*GCLM*) cells, compared to siCTL (*Figure 3o*). Collectively, these data suggest that PGC-1s play an important role in promoting resistance to doxorubicin and epirubicin by regulating the expression of genes that contribute to resistance to each drug.

## Doxorubicin-resistant cells rely on glutamine for sustained ATP production

Given that glutamine is a key fuel for cancer cell growth (*McGuirk et al., 2013*) and that DoxR cells use glutamine for glutathione synthesis, we hypothesized that anthracycline-resistant cells may be particularly sensitive to glutamine withdrawal. The proliferation rates of all cell lines were severely affected by glutamine withdrawal. However, while Control cells could maintain a low level of proliferation, both DoxR and EpiR cells were unable to proliferate in the absence of glutamine (*Figure 4a*). This was accompanied by a small, but significant, increase in cell death, by 5% in DoxR and 10% in EpiR cells, compared to Control cells (*Figure 4b*). Comparatively, neither Control, DoxR, nor EpiR cells were significantly affected by glucose limitation (*Figure 4—figure supplement 1a*).

To further quantify the impact of glutamine withdrawal on cellular metabolism, we assessed the bioenergetics of resistant cells in the presence or absence of glutamine, through extrapolation of $J_{ATP}$ rates from OCR and ECAR measurements (*Figure 4c*, *Figure 4—figure supplement 1b*). After 4 hr of glutamine withdrawal, all three cell lines showed a decrease in mitochondrial ATP production, with DoxR cells being the most affected (*Figure 4d*). This decrease in mitochondrial ATP production was compensated for by an increase in ATP production through glycolysis in EpiR and Control cells, but not DoxR cells (*Figure 4e*). This compensatory increase in glycolytic ATP production was sufficient to maintain total ATP production (mitochondria and glycolysis) in Control and EpiR cells, while the lack of compensatory glycolytic ATP production in DoxR cells led to a drop in their total ATP production (*Figure 4f*). The fact that DoxR cells were unable to increase glycolytic ATP production to compensate for diminished mitochondrial ATP production upon glutamine withdrawal may indicate a defect in glycolytic regulation or ATP sensing (*Figure 4e–f*).

We also measured the maximal mitochondrial bioenergetic capacity as well as the maximal glycolytic capacity of these cells (*Figure 4h–j*; *Mookerjee et al., 2017*). Strikingly, glutamine starvation

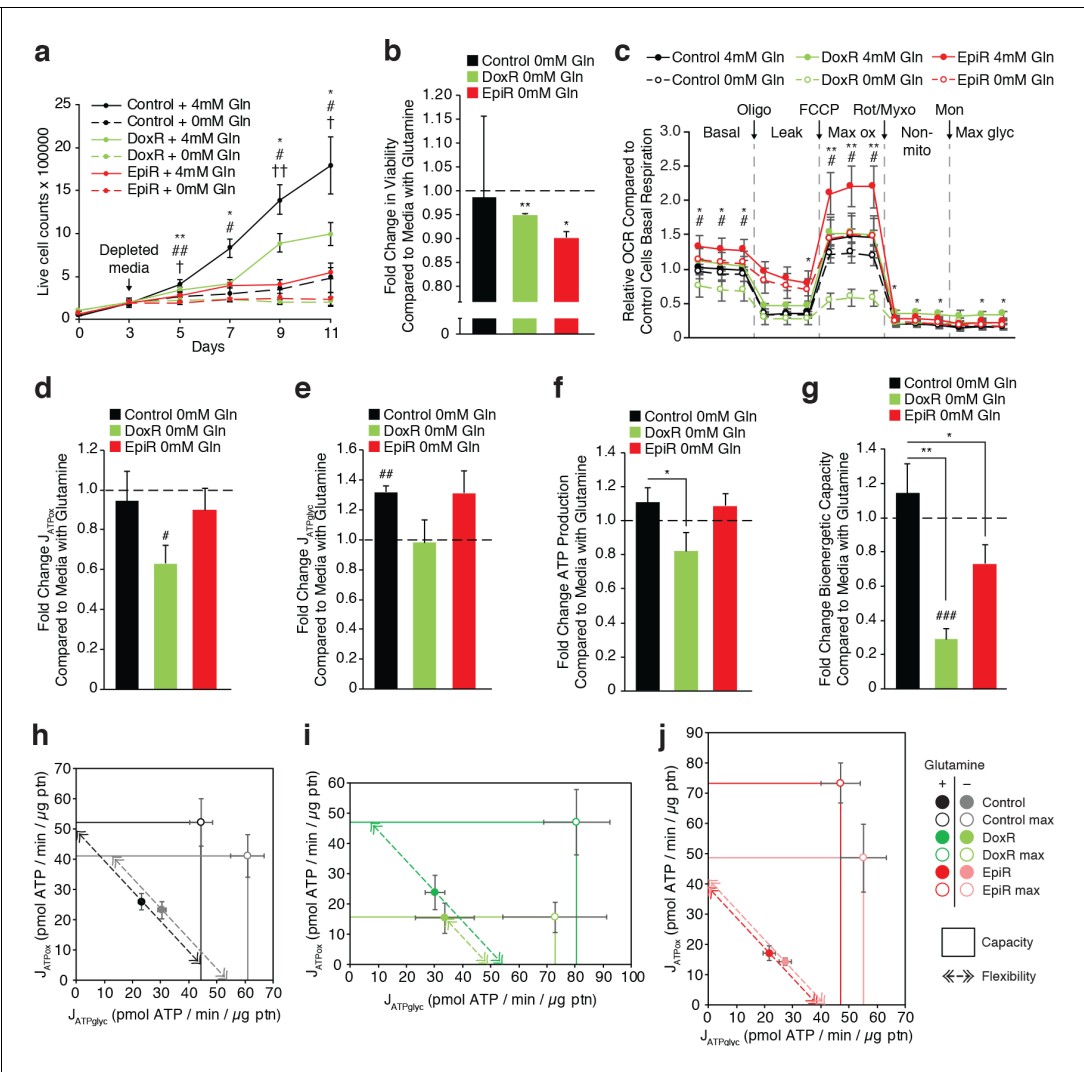

**Figure 4.** Doxorubicin-resistant breast cancer cells are reliant on glutamine to sustain their OXPHOS bioenergetic capacity. (a) Growth of Control, DoxR, and EpiR cells in glutamine-rich (4 mM) or glutamine-deprived (0 mM) conditions. Cells were differentially seeded to reach a total of 200,000 cells at day 3 under normal growth conditions; media was then changed for glutamine-rich or glutamine-depleted media, daily (day 3–11). N = 3, *p<0.05 **p<0.01 Control cells 0 mM vs 4 mM, #p<0.05 ##p<0.01 DoxR cells 0 mM vs 4 mM, †p<0.05 ††p<0.001 EpiR cells 0 mM vs 4 mM (paired Student's t-test). (b) Fold change in viability of Control, DoxR, and EpiR cells after 6 days glutamine withdrawal compared to glutamine-rich (4 mM) conditions. N = 3, *p<0.05 **p<0.01 0 mM vs 4 mM glutamine (paired Student's t-test). (c) Seahorse analysis of basal, leak (oligomycin), maximum (FCCP), and non-mitochondrial (rotenone and myxothiazol) oxygen consumption rates, as well as after addition of monensin, in Control, DoxR, and EpiR cells after 4 hr glutamine withdrawal compared to glutamine-rich (4 mM) conditions, presented as relative to the basal respiration rate of Control cells in the presence of glutamine. N = 5, *p<0.05 **p<0.01 DoxR 0 mM Gln vs DoxR 4 mM Gln, #p<0.05 EpiR 0 mM Gln vs EpiR 4 mM Gln (paired Student's t-test). (d) Fold change in ATP production by oxidative phosphorylation ($J_{ATPox}$) of Control, DoxR, and EpiR cells after 4 hr glutamine withdrawal compared to glutamine-rich (4 mM) conditions. N = 5, #p<0.05 0 mM vs 4 mM (paired Student's t-test). (e) Fold change in ATP production from glycolysis ($J_{ATPglyc}$) of Control, DoxR, and EpiR cells after 4 hr glutamine withdrawal compared to glutamine-rich (4 mM) conditions. N = 5, ##p<0.01 0 mM vs 4 mM (paired Student's t-test). (f) Fold change in total ATP production rate ($J_{ATP}$) of Control, DoxR, and EpiR cells after 4 hr glutamine withdrawal compared to glutamine-rich (4 mM) conditions. N = 5, *p<0.05 resistant vs Control cells (paired Student's t-test). (g) Fold change in bioenergetic capacity of Control, DoxR, and EpiR cells after 4 hr glutamine withdrawal compared to glutamine-rich (4 mM) conditions. N = 5, *p<0.05 **p<0.01 resistant vs Control cells, ###p<0.001 0 mM vs 4 mM glutamine. (h-j) Bioenergetic space plots of Control (h), DoxR (i), and EpiR (j) cells after 4 hr glutamine withdrawal compared to glutamine-rich (4 mM) conditions. Solid points represent actual $J_{ATPglyc}$ and $J_{ATPox}$ values, hollow points represent theoretical maximums for $J_{ATPox}$ (FCCP) and $J_{ATPglyc}$ (rotenone, myxothiazol, monensin). Dotted line arrows' length represents flexibility of ATP production within maximum boundaries (solid lines). Area under maximum boundaries represents the bioenergetic capacity. N = 5. All data presented as averages ± S.E.M.

The online version of this article includes the following figure supplement(s) for figure 4:

**Figure supplement 1.** Proliferation and bioenergetics of Control, DoxR, and EpiR cells under nutrient deprivation conditions.

entirely abrogated the spare oxidative capacity of DoxR cells (*Figure 4i*, box height relative to basal point). This significantly reduced the total bioenergetic capacity (mitochondrial and glycolytic capacities combined, *Figure 4i* box area, *Figure 4g*) of DoxR cells and their bioenergetic flexibility—the ability to dynamically shift glycolysis and OXPHOS rates while maintaining a constant ATP production rate (*Figure 4i*, dotted arrow length). Control and EpiR cells showed diminished oxidative capacity (box height) and increased glycolytic capacity (box width) as well as no change in flexibility after glutamine withdrawal (*Figure 4h,j*). However, the decrease in oxidative capacity in EpiR cells was greater than their increase in glycolytic capacity, leading to a reduced total bioenergetic capacity in the absence of glutamine (*Figure 4g,j*). The total bioenergetic capacity of EpiR cells remained nevertheless greater than that of DoxR cells (*Figure 4g*). Taken together, these data show that glutamine is specifically important for doxorubicin resistant breast cancer cells, not only for glutathione synthesis but also for mitochondrial ATP production.

## Independently-derived models of doxorubicin- and epirubicin-resistance confirm their respective dependance on glutathione metabolism and oxidative phosphorylation

To independently confirm the distinct metabolic adaptations to doxorubicin and epirubicin of the well-characterized cells in this study (*Hembruff et al., 2008*), we generated new resistant cell lines from drug-naive MCF-7 cells over the course of 8 months, following a similar step-wise process and up to a common end-point dose of 100 nM doxorubicin (D100 cells) or epirubicin (E100 cells; *Figure 5a*). Matched parental control (Ctl) cells were maintained in 0.1% DMSO through parallel passages (*Figure 5a*). Importantly, while we sought to confirm the distinct metabolic adaptations to each drug, it was expected that there would be some variability between these independently derived resistant cells and the cells used in the rest of study given that cancer cells may engage different adaptation strategies to develop resistance to chemotherapy (*Edwardson et al., 2013*).

As seen in DoxR and EpiR cells (*Figure 2*), key oxidative response genes (*NFE2L2*, *NQO1*) as well as *GLS* and *IDH1* were significantly overexpressed in both D100 and E100 cells, while only D100 cells exhibited significant overexpression of glutathione metabolism genes (*GCLC*, *GCLM*, *GSS*, *GSR*) compared to parental control (Ctl) cells (*Figure 5b*). Accordingly, stable isotope tracing analyses confirmed that D100 have significantly increased *de novo* synthesis of glutathione from glutamine compared to Ctl cells, whereas it was significantly decreased in E100 cells (*Figure 5c* and *Figure 5—figure supplement 1a*). D100 cells also displayed a lower ROS signal than E100 cells, both at baseline (*Figure 5d*) and after $H_2O_2$ treatment (*Figure 5e*), while both resistant cells had lower ROS signals than Ctl cells.

Similar to EpiR cells (*Figure 3*), E100 cells also had significantly higher expression of *PPARGC1A* (*Figure 5b*), along with increased oxygen consumption and extracellular acidification rates, increased total ATP production rates, increased basal and maximum oxidative ATP production rates ($J_{ATPox}$), as well as greater bioenergetic capacity compared with Ctl cells (*Figure 5—figure supplement 1b,c* and *Figure 5f–i*). Interestingly, and in contrast to DoxR cells, D100 cells displayed a significant increase in the expression of *PPARGC1A* (*Figure 5b*), as well as elevated maximum oxidative ATP production rate and bioenergetic capacity compared to Ctl (*Figure 5g–i*). Despite these differences in bioenergetics between the two cell models of doxorubicin resistance, these results confirm that epirubicin resistant cells display higher oxidative capacity than doxorubicin-resistant cells (*Figure 5g–h*).

Overall, these independently derived cell models of doxorubicin and epirubicin resistance broadly replicate the central findings of the manuscript, namely that doxorubicin-resistant cells have an elevated usage of glutamine for glutathione synthesis and that epirubicin-resistant cells display markedly increased OXPHOS capacity. Furthermore, given that the independently derived cell lines were selected to a common end-point dose of 100 nM of doxorubicin or epirubicin, these data further demonstrate that these metabolic adaptations are specific to the drug and not the dose.

## Tailored metabolic adaptations underpinning resistance to doxorubicin and epirubicin lead to primary actionable vulnerabilities

Despite some similar mechanisms supporting resistance to doxorubicin and epirubicin in breast cancer cells, our results have thus far shown that drug-dependent dominant metabolic adaptations arise

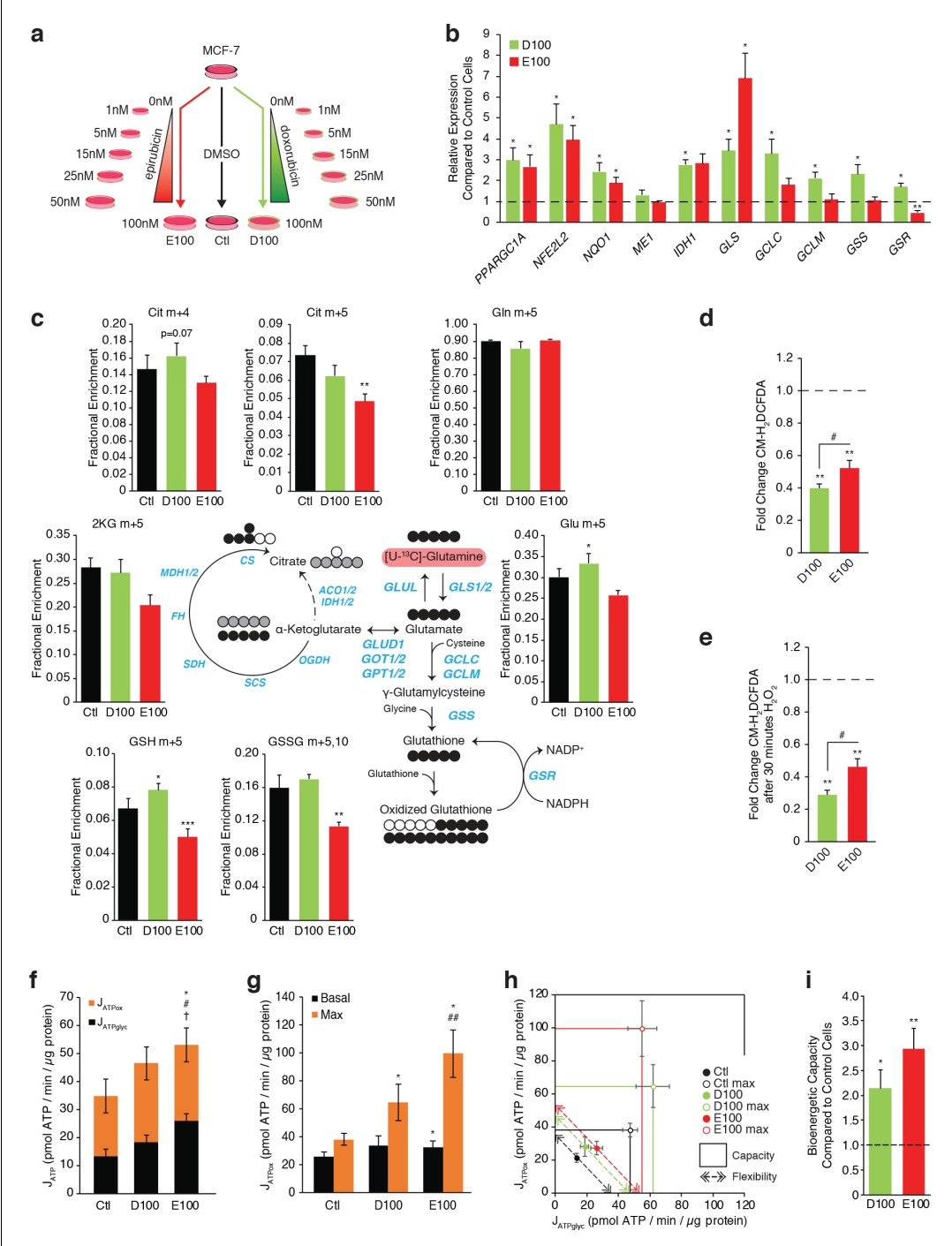

**Figure 5.** Independently derived resistant models confirm specific metabolic adaptations to doxorubicin and epirubicin. (a) Model detailing generation of breast cancer cells resistant to increasing concentrations of anthracyclines, to a final stable extracellular concentration of 100 nM of doxorubicin (D100) or epirubicin (E100). Control cells (Ctl) were maintained in DMSO in parallel passages. (b) Relative expression of *PPARGC1A* and selected metabolic, glutathione, and oxidative response genes in D100 and E100 compared to Ctl. N = 3–6. *p<0.05 resistant vs Ctl cells (paired Student's t-test). (c) [U-$^{13}$C]-Glutamine tracing to glutamate and into the citric cycle (2 KG m + 5, citrate m + 4 and m + 5, malate m + 3, 2 hr tracer) or through to glutathione (GSH m + 5 and GSSG m + 5,10, 4 hr tracer) in Ctl, D100, and E100 expressed as fractional enrichment. N = 3–5, *p<0.05 **p<0.01 ***p<0.001 resistant vs Ctl cells (paired Student's t-test). (d) Fold change of ROS signal in D100 and E100 cells compared to Ctl cells, detected by CM-H$_2$DCFDA staining. N = 5, **p<0.01 resistant vs Control cells, #p<0.05 D100 vs E100 (paired Student's t-test). (e) Fold change of ROS signal in D100 and E100 compared to Ctl cells, after 30 min treatment with 0.03% H$_2$O$_2$. N = 5, **p<0.01 resistant vs Control cells, #p<0.05 D100 vs E100 (paired Student's t-test). (f) Quantification of ATP production (J$_{ATP}$) from oxidative phosphorylation (J$_{ATPox}$) or glycolysis (J$_{ATPglyc}$) in Ctl, D100, and E100. N = 5, *p<0.05

*Figure 5 continued on next page*

Figure 5 continued

$J_{ATPox}$ #p<0.05 $J_{ATPglyc}$ †p<0.05 $J_{ATP}$, resistant vs Ctl cells (paired Student's t-test). (g) Quantification of ATP production by OXPHOS ($J_{ATPox}$) in Ctl, D100 and E100 cells under basal conditions and at peak bioenergetic capacity (Max). N = 5, *p<0.05 resistant vs Ctl cells, ##p<0.01 E100 vs D100 (paired Student's t-test). (h) Bioenergetic space plots of Ctl, D100, and E100. Solid points represent actual $J_{ATPglyc}$ and $J_{ATPox}$ values, hollow points represent maximums for $J_{ATPox}$ (FCCP) and $J_{ATPglyc}$ (rotenone, myxothiazol, monensin). Length of dotted line arrows represents flexibility of ATP production within maximum boundaries (solid lines). Area under maximum boundaries represents the bioenergetic capacity. N = 5. (i) Fold change in bioenergetic capacity of D100 and E100 compared to Ctl. N = 5 *p<0.05 **p<0.01 resistant vs Ctl cells (paired Student's t-test). All data presented as averages ± S.E.M.

The online version of this article includes the following figure supplement(s) for figure 5:

**Figure supplement 1.** Supporting information for independently derived resistant models.

in resistant cells. Indeed, resistance to doxorubicin is linked to glutathione metabolism, whereas resistance to epirubicin is tied to enhanced mitochondrial bioenergetic capacity. To further demonstrate the importance of these tailored metabolic adaptations, we sought to determine whether these primary vulnerabilities are targetable.

Using the biguanide phenformin, we assessed if epirubicin-resistant cells are specifically sensitive to inhibition of OXPHOS. As expected, phenformin had a strong and dose-dependent effect on the growth of drug-naïve (Control and Ctl) breast cancer cells (*Figure 6a,b*). Epirubicin-resistant cells (EpiR and E100) were similarly vulnerable to phenformin in a dose-dependent manner, whereas doxorubicin-resistant (DoxR and D100) cells were only mildly responsive to the drug; both epirubicin-resistant cells were significantly more sensitive to phenformin than their doxorubicin-resistant counterparts (*Figure 6a,b*).

Next, given that doxorubicin-resistant cells display elevated glutathione metabolism and that they rely on glutamine to fuel glutathione synthesis, we explored this pathway for therapeutic intervention through targeted therapy with buthionine sulfoximine (BSO), an inhibitor of the catalytic subunit of glutamate-cysteine ligase (GCLC) (*Drew and Miners, 1984*). Both doxorubicin-resistant cell models (DoxR and D100) were acutely and specifically sensitive to BSO treatment *in vitro* (*Figure 6c, d*). BSO was highly effective in reducing proliferation of DoxR cells even at the lowest dose tested (50 μM, 60% reduction in viable cell count), while having little to no effect on Control and EpiR cells at that concentration (*Figure 6c*). These results were further replicated in D100 cells, which were significantly more sensitive to BSO than Ctl and E100 cells (*Figure 6d*).

Given the potency of BSO treatment *in vitro*, we explored its effectiveness *in vivo* by injecting DoxR and EpiR cells into opposing mammary fat pads of immunocompromised mice, supplemented with subcutaneous estrogen in order to promote tumor growth. Once DoxR tumors reached ~100 mm³, mice were divided into two groups and treated daily by intraperitoneal injection with either 450 mg/kg of BSO or with vehicle (PBS), for 20 days (*Figure 6e*). While all EpiR tumors grew to a larger size than DoxR tumors before the start of treatment, both tumor types similarly doubled in size over the 20 days when treated with vehicle (from 100 mm³ to 200 mm³ for DoxR, from 325 mm³ to 650 mm³ for EpiR, *Figure 6f,g*). Daily BSO treatment significantly reduced the growth of DoxR tumors, which only increased in size by a factor of 30%, while the growth of EpiR tumors were unaffected (*Figure 6f,g*). Tumor excision after 20 days of treatment further confirmed this result (*Figure 6e*). Collectively, these data highlight that despite common adaptations to chemotherapeutics, distinct primary metabolic vulnerabilities arise in doxorubicin and epirubicin resistance, which can be targeted through metabolic interventions to impair drug-resistant tumor growth (*Figure 6h*).

## Discussion

Whereas most studies on therapeutic resistance have focused on single agents or multi-drug resistance, our study presents an unprecedented side-by-side comparison of the metabolic adaptations driving resistance to distinct therapeutic agents within the same drug class. Here, we show that two anthracycline drugs, doxorubicin and epirubicin, elicit different actionable primary metabolic adaptations that support therapeutic resistance and breast cancer cell survival.

Specifically, our findings indicate that doxorubicin-resistant, but not epirubicin-resistant, cells rely on elevated usage of glutamine for *de novo* glutathione synthesis. This metabolic dependency can be targeted, as demonstrated by the fact that doxorubicin-resistant cells and tumors are significantly

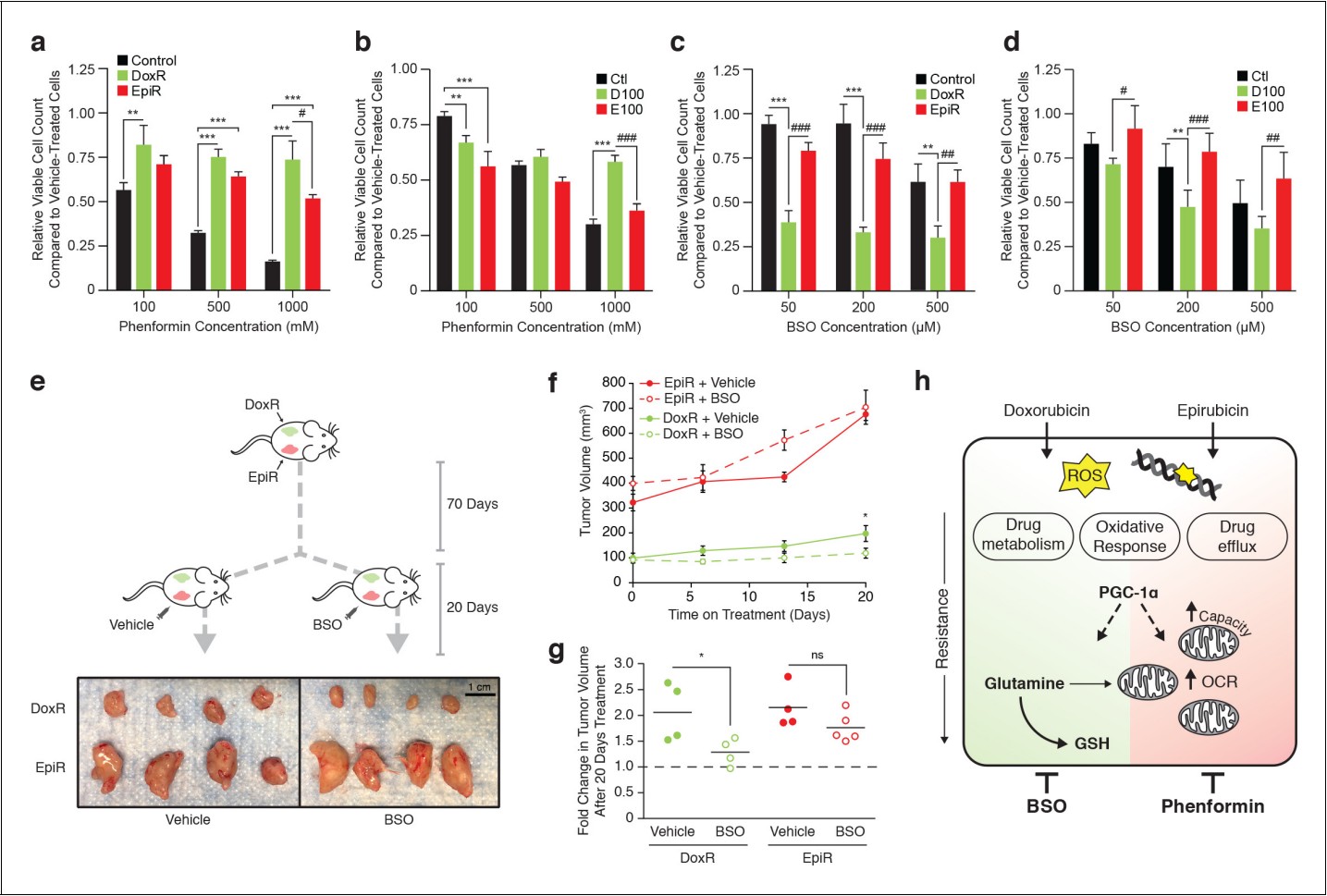

**Figure 6.** Tailored metabolic adaptations underpinning resistance to doxorubicin and epirubicin lead to primary actionable vulnerabilities *in vitro* and *in vivo*. (**a**) Relative viable cell count of Control, DoxR, and EpiR cells after 3 days treatment with a combination of phenformin and their respective drug (DMSO, 98.1 nM doxorubicin, or 852 nM epirubicin). Data are shown as relative viable cell count of phenformin-treated cells compared to cells treated with vehicle (water). N = 4, *p<0.05 **p<0.01 ***p<0.001 resistant vs Control cells, #p<0.05 DoxR vs EpiR cells (paired Student's t-test). (**b**) Relative viable cell count of Ctl, D100, and E100 cells after 3 days treatment with a combination of phenformin and their respective drug (DMSO, 100 nM doxorubicin, or 100 nM epirubicin). Data are shown as relative viable cell count of phenformin-treated cells compared to cells treated with vehicle (water). N = 4, **p<0.01 ***p<0.001 resistant vs Ctl cells, ***p<0.001 D100 vs E100 cells (paired Student's t-test). (**c**) Relative viable cell count of Control, DoxR, and EpiR cells after 3 days treatment with a combination of buthionine sulfoximine (BSO) and their respective drug (DMSO, 98.1 nM doxorubicin, or 852 nM epirubicin) after 7 weeks of drug holiday. Data are shown as relative viable cell count of BSO-treated cells compared to cells treated with vehicle (water). N = 3, **p<0.01 ***p<0.001 resistant vs Control cells, ##p<0.01 ###p<0.001 DoxR vs EpiR cells (paired Student's t-test). (**d**) Relative viable cell count of Ctl, D100, and E100 cells after 3 days treatment with a combination of BSO and their respective drug (DMSO, 100 nM doxorubicin, or 100 nM epirubicin). Data are shown as relative viable cell count of BSO-treated cells compared to cells treated with vehicle (water). N = 3, **p<0.01 resistant vs Control cells, #p<0.05 ##p<0.01 ###p<0.001 D100 vs E100 cells (paired Student's t-test). (**e**) Picture of end-point DoxR and EpiR tumors after 70 days of growth in the opposing mammary fat pads of NOD Scid Gamma mice supplemented twice weekly with subcutaneous injection of 5 μg estrogen, followed by 20 days with daily intraperitoneal injection of either 450 mg/kg BSO or vehicle (PBS). (**f**) Volume of DoxR and EpiR tumors measured over 20 days with daily treatments of either 450 mg/kg BSO or vehicle (PBS) by intraperitoneal injection. Data are shown as tumor volume on day 0, 6, 13, and 20. N = 4–5, *p<0.05 BSO vs vehicle (two-way ANOVA, Sidak's post-hoc test). (**g**) Fold change in DoxR and EpiR tumor volumes after 20 days of daily treatment with either 450 mg/kg BSO or vehicle (PBS) by intraperitoneal injection. Data are shown as fold changes for individual tumors, relative to baseline tumor volume at day 0 (dotted line). The average fold change for each condition is shown by horizontal lines. N = 4–5, *p<0.05 BSO vs vehicle (Student's t-test). (**h**) Schematic of common adaptation mechanisms and distinct primary metabolic dependencies in anthracycline resistant breast cancer cells. Doxorubicin and epirubicin both induce production of reactive oxygen species (ROS) and intercalate nucleic acids and inhibit topoisomerase II, leading to double-strand DNA breaks. Both doxorubicin- and epirubicin-resistant cells engage oxidative response, drug metabolism, and drug efflux pathways to overcome these drug mechanisms, and both are dependent on expression of PGC-1α for their survival. PGC-1α-regulated pathways may further underpin distinct and context-dependent metabolic adaptations to either drug. Compared to drug-sensitive control cells, doxorubicin-resistant cells rely on glutamine for *de novo* glutathione (GSH) synthesis and for mitochondrial ATP production, while epirubicin-resistant cells display elevated mitochondrial content, oxygen consumption rate (OCR), and oxidative bioenergetic capacity. These distinct

*Figure 6 continued on next page*

*Figure 6 continued*

primary metabolic dependencies are actionable, as epiribucin-resistant cells are more sensitive to phenformin treatment than doxorubicin-resistant cells, and the latter are specifically sensitive to inhibition of glutathione synthesis by buthionine sulfoximine (BSO) both *in vitro* and *in vivo*. Unless otherwise noted, all data presented as averages ± S.E.M.

more sensitive than epirubicin-resistant cells to therapeutic intervention with BSO, which interferes with glutathione synthesis. Given that side effects such as cardiotoxicity limit the use of anthracyclines in patients to a restrictive cumulative total lifetime dose, there is important clinical relevance in reducing tumor growth in anthracycline-resistant patients through a secondary treatment option such as BSO without administrating additional anthracycline chemotherapy (*Barrett-Lee et al., 2009*). In contrast to doxorubicin-resistant cells, epirubicin-resistant cells display a drastic increase in OXPHOS and oxidative bioenergetic capacity and were more sensitive than doxorubicin-resistant cells to treatment with phenformin. This aligns with the compendium of evidence showing that dependence on mitochondrial energy metabolism and oxidative phosphorylation is a widespread characteristic of drug resistance, across several cancer types and therapeutic interventions (*Caro et al., 2012*; *Bosc et al., 2017*; *Matassa et al., 2016*; *Viale et al., 2014*; *Vellinga et al., 2015*; *Ippolito et al., 2016*; *Vazquez et al., 2013*; *Lee et al., 2017*; *Farge et al., 2017*; *Kuntz et al., 2017*; *Thompson et al., 2017*).

Our findings are particularly interesting given the structural similarity of doxorubicin and epirubicin. In line with this, breast cancer cells resistant to either drug displayed well-known mechanisms of resistance to anthracyclines, including increased drug efflux, lysosomal activity, and oxidative stress response (*Hembruff et al., 2008*; *Guo et al., 2016*). Their different metabolic state may, in part, be due to minor structural differences between the two drugs, leading to distinct on- and off-target effects under sustained treatments with either drug (*Salvatorelli et al., 2006*). For example, while cardiotoxicity is a common side effect of sustained anthracycline treatment, epirubicin has been shown to induce less cardiotoxic effects than doxorubicin, even if both drugs display equivalent response rate to treat breast cancer (*Mao et al., 2019*). Given that cardiotoxicity is linked to oxidative stress, it is possible that breast cancer cells treated with doxorubicin may face a greater oxidative challenge over the course of treatment than those treated with epirubicin, which aligns with a greater dependence of doxorubicin-resistant cells on *de novo* glutathione synthesis (*Salvatorelli et al., 2006*). Accordingly, doxorubicin-resistant cells also displayed much greater engagement of oxidative stress response than epirubicin-resistant cells. Interestingly, epirubicin-resistant cells displayed an elevated level of uncoupled respiration, which may represent an alternate approach to minimizing ROS production in this model (*Echtay et al., 2002*; *Brand, 2000*). Indeed, targeting uncoupling proteins has previously been shown to sensitize multi-drug-resistant leukemia cells to both doxorubicin and epirubicin (*Mailloux et al., 2010*).

Crucially, our study further upholds the viability of exploiting metabolic alterations associated with resistance to chemotherapeutic drugs to increase their success rate (*Zaal and Berkers, 2018*). This strategy has already shown success in numerous cancers; for example, inhibition of amino acid recycling sensitized neuroblastomas to cisplatin (*Gunda et al., 2020*), fueling histidine catabolism via histidine supplementation increases sensitivity of leukemic xenografts to methotrexate (*Kanarek et al., 2018*), and the glutaminase inhibitor CB-839 synergistically enhances the cytotoxicity of carfilzomib in treatment-resistant multiple myeloma cells, notably through its inhibition of glutamine-fueled mitochondrial respiration (*Thompson et al., 2017*).

It is also notable that the master regulator of mitochondrial metabolism PGC-1α regulates a significant number of pathways implicated in therapy resistance, including OXPHOS (*Mootha et al., 2003*), oxidative stress response (*St-Pierre et al., 2006*), glutamine metabolism (*McGuirk et al., 2013*), and glutathione metabolism (*Guo et al., 2018*). The context-dependent roles of PGC-1α may therefore underpin specific metabolic vulnerabilities in both doxorubicin and epirubicin resistance in breast cancer. Accordingly, both doxorubicin- and epirubicin-resistant cells were sensitive to PGC-1α knockdown in our study. This aligns with the emerging role of PGC-1α in driving bioenergetic flexibility and metabolic plasticity in the face of survival challenges involved in cancer progression (*Tan et al., 2016*; *McGuirk et al., 2020*; *Andrzejewski et al., 2017*); advanced cancers need to be adaptable, and thereby the context-dependent adaptations conferred by PGC-1α could further contribute to the difficulty in treating advanced cancers. Indeed, similar to OXPHOS and mitochondrial

energy metabolism, PGC-1α has been shown to be implicated in drug resistance across cancer types and through various mechanisms (see *Supplementary file 1*). While therapeutically targeting transcription factors that relay PGC-1α effects may represent an effective strategy in some cases (*De Vitto et al., 2019*; *Deblois et al., 2016*), attempts to directly target PGC-1α have unfortunately shown little success thus far.

Ultimately, targeting global regulators of metabolic plasticity like PGC-1α may be promising as a broad strategy for treatment of therapeutic-resistant cancers. However, targeted interventions exploiting the primary metabolic dependencies associated to specific resistant cancers—such as using BSO as a therapeutic intervention for doxorubicin-resistant breast cancer—may represent a more immediate and effective approach.

## Materials and methods

### Tissue culture and generation of stable cell lines

MCF-7$_{CC}$, MCF-7$_{DOX-2}$, and MCF-7$_{EPI}$ cells were obtained from Dr. Amadeo Parissenti (*Hembruff et al., 2008*). Briefly, MCF-7$_{DOX-2}$ and MCF-7$_{EPI}$ were selected over 12 sequential dose increases with their respective anthracycline drug (doxorubicin, epirubicin) to maximal doses of 98.1 nM and 852 nM (*Hembruff et al., 2008*). MCF-7$_{CC}$ cells were maintained in 0.1% DMSO through parallel passages (*Hembruff et al., 2008*). For simplicity, MCF-7$_{CC}$, MCF-7$_{DOX-2}$, and MCF-7$_{EPI}$ cells are referred to only as Control, DoxR, and EpiR in this study. Cells were cultured in high-glucose Dulbecco's Modified Eagle's Medium (DMEM, Wisent #319–005 CL), 10% FBS, and penicillin/streptomycin at 37°C and 5% CO$_2$. New resistant MCF-7 models were derived from MCF-7 cells obtained from the American Type Culture Collection (ATCC) and cultured in similar media under increasing doses of doxorubicin (Abmole Biosciences #M1969) or epirubicin (Sigma Aldrich #E9406) from 0.1 nM to a final dose of 100 nM, over the course of 8 months. Ctl cells were maintained in 0.1% DMSO through parallel passages. All cells were maintained in culture with a constant dose of their respective drug or DMSO control, at all times unless otherwise specified.

### Proliferation and viability

Proliferation assays were performed by seeding 200,000 cells in 35 mm plates and growing in full media as described above. For glutamine withdrawal experiments, media was replaced on day three with glutamine-free media or glutamine-free media re-supplemented with 4 mM glutamine. To determine cell counts, cells were washed, trypsinized, and counted using a TC10 automated cell counter (Bio-Rad). Viability was determined by exclusion of trypan blue dye.

### Mouse experiments

Four million DoxR or EpiR cells were injected into opposing mammary fat pads of NOD Scid Gamma mice supplemented twice weekly with subcutaneous injection of 5 micrograms of estrogen in corn oil. Seventy days after tumor cell injection mice were divided into two groups and treated daily by intraperitoneal injection with either 450 mg/kg of L-Buthionine-sulfoximine (Sigma Aldrich #B2515) or vehicle (PBS). Tumor volume was measured weekly using caliper measurements and the formula $length * width^2 * \frac{\pi}{6}$.

### Gene expression

Total RNA from cultured cells was extracted using the Aurum Total RNA Mini Kit (Bio-Rad, Mississauga, Canada) and was reverse transcribed with iScript cDNA Synthesis kit (Bio-Rad). mRNA expression analyses by real-time PCR were performed using iQ SYBR Green Supermix (Bio-Rad) and gene-specific primers with the MyiQ2 Real-Time Detection System (Bio-Rad). Values were normalized to TATA-binding protein (*TBP*) expression. All primer sequences are listed in *Supplementary file 2*.

### Gene expression profiling, enrichment analyses, and ranked gene list comparisons

Gene expression profiling of Control, DoxR, and EpiR cells was performed with Genome Québec (Montreal, Canada) using the Affymetrix Human Gene 2.0 ST Array (HT) system, for which RNA was isolated as described above. The .CEL files were analyzed and pre-processed using the Affymetrix

Transcriptome Analysis Console software (RRID:SCR_016519). These data have been deposited in NCBI's Gene Expression Omnibus (RRID:SCR_005012, *Edgar et al., 2002*) and are accessible through GEO Series accession number GSE125187 (https://www.ncbi.nlm.nih.gov/geo/query/acc.cgi?acc=GSE125187). Gene Set Enrichment Analysis (GSEA, RRID:SCR_003199) was performed on ranked gene lists, where ranks were designated by the sign of the fold change multiplied by the logarithm of the p-value (*Subramanian et al., 2005*).

To compare with patient data, differential expression analyses of this microarray and a publicly available patient dataset (GEO accession GSE43816, *Gruosso et al., 2016*) were performed using the R (RRID:SCR_001905) Bioconductor (RRID:SCR_006442) package 'LIMMA' and lists were ranked by t-test statistics (*R Development Core Team, 2019*; *Phipson et al., 2016*). There are 34,744 and 20,474 genes in the cell line and GEO data set, respectively. 19,038 genes appeared in both data sets. To compare the observed size of overlap between two ordered gene lists to the expected overlap when two lists are independent, we followed the methods outlined by *Yang et al., 2006*. Specifically, we measured the expected overlap by randomly shuffling the rank order of one list and measuring the size of overlap, repeating this over 1000 permutations. The R Bioconductor (RRID:SCR_006442) package 'OrderedList' was used to calculate the expected overlap.

## shRNA screen for drug target genes

A list of 1215 genes related to clinically-approved drugs was generated based on DrugBank and The NCGC Pharmaceutical Collection (*Huang et al., 2011*). A library with 7847 shRNAs targeting these genes (FDA library) was constructed from the arrayed and sequence-verified RNAi Consortium (TRC) human genome-wide shRNA collection, provided by The McGill Platform for Cell Perturbation (MPCP) of the Rosalind and Morris Goodman Cancer Research Centre and Biochemistry Department at McGill University. This druggable library consists of 11 plasmid pools. Lentiviral supernatants were generated as described at http://www.broadinstitute.org/rnai/public/resources/protocols. DoxR and EpiR cells were infected separately by the 11 virus pools. Cells were then pooled and plated at 500,000 cells per 15 cm dish with 1000 times of coverage in presence of doxorubicin or epirubicin (respectively), for a total of 32 dishes per cell line. Genomic DNA was extracted from the remaining cells in the original pool, as well as in a pool of all 32 dishes after 7 days of growth, and sequencing libraries were built as previously described (*Huang et al., 2012*). shRNA stem sequence was segregated from each sequencing reads and aligned to TRC library. The matched reads were counted, normalized, and analyzed in R (RRID:SCR_001905) using MAGeCK (v0.5.5) (*Li et al., 2014*). Hits were ranked by p-value from most depleted to most enriched in DoxR or EpiR after 7 days, and ranked lists were further analyzed for over-represented pathways using Gene Set Enrichment Analysis (RRID:SCR_003199, *Mootha et al., 2003*; *Subramanian et al., 2005*).

## Stable isotope tracer analysis

Stable isotope tracer analyses (SITA) were performed in GC/MS as previously described (*McGuirk et al., 2013*). Briefly, cells were seeded in 6-well dishes to achieve 70–80% confluency after 48 hr. Media was then replaced by DMEM without glucose, sodium pyruvate or L-glutamine (Wisent #319–062) supplemented with 10% dialyzed FBS, 25 mM glucose, 1X sodium pyruvate, and 4 mM glutamine for 2 hr to equilibrate metabolism. Media was further changed to equivalent labeled media made with either 25 mM [U-$^{13}$C]-glucose or 4 mM [U-$^{13}$C]-glutamine for the indicated time points. DMSO, doxorubicin, or epirubicin were present in the media throughout. Cells were washed twice with saline at 4°C, quenched in 80% HPLC-grade methanol at −80°C, sonicated, and centrifuged. Supernatants were supplemented with internal control (750 ng myristic acid-D$_{27}$) and dried in a cold trap overnight (Labconco) at −1°C. Pellets were solubilized in 10 mg/mL methoxyamine-HCl in pyridine, incubated 30 min at 70°C, and derivatized with N-tert-Butyldimethylsilyl-N-methyltrifluoroacetamide (MTBSTFA) for 1 hr at 70°C. 1 μL was injected into an Agilent 5975C GC/MS in SCAN mode and analyzed using Chemstation (Agilent Techologies, RRID:SCR_015742) and Masshunter softwares (Agilent Technologies, RRID:SCR_015040).

Tracing glutamine carbons to glutathione was done using a similar labeling method as above. Cells were washed twice with 150 mM ammonium formate buffer in HPLC water at 4°C, quenched in 50% HPLC-grade methanol at −20°C on dry ice, and phase-separated using acetonitrile, water, and dichloromethane after vigorous bead-beating and vortexing. The aqueous phase was collected and

dried in a cold trap overnight at −1°C. Pellets were solubilized in HPLC water and 5 µL was injected into an Agilent 6540 UHD Accurate-Mass Q-TOF LC/MS system coupled to ultra-high pressure liquid chromatography (UHPLC, 1290 Infinity LC System) and analyzed using Masshunter software.

All isotopic corrections were performed using an in-house algorithm designed by SM as previously described (*McGuirk et al., 2013*).

## Metabolomics

Steady-state metabolite abundances were determined using GC/MS and LC/MS systems, using unlabeled media. Citric acid cycle, glycolytic intermediates, and fatty acids were measured in GC/MS as described above. Amino acids were measured in a Q-TOF system as described above. Nucleotide abundances were determined by washing 70–80% confluent 10 cm plates of cells with 150 mM ammonium formate at 4°C, quenched in 80% HPLC-grade methanol at −80°C on dry ice, after which the cell slurry was quickly transferred to tubes equilibrated in liquid nitrogen. After 24 hr, these were phase-separated using water and dichloromethane after vigorous bead-beating and vortexing. The aqueous phase was collected and flash-frozen in liquid nitrogen, then dried in a cold trap at −1°C. Once dry, pellets were maintained at −80°C and solubilized in HPLC water immediately before injection into an Agilent 6430 Triple Quadrupole LC/MS system coupled to ultra-high pressure liquid chromatography (UHPLC, 1290 Infinity LC System) separation for fast targeted analysis.

Glutathione levels were quantified using an Agilent 1100 series HPLC (*Mailloux et al., 2014*). Three days post-seeding, cells grown in 6-well plates were washed twice with ice-cold PBS, flash-frozen on dry ice and kept at −80°C until further processing. Cells from parallel plates were counted for normalization. Cells were lysed on ice for 20 min using a mix of 125 mM sucrose, 1.5 mM EDTA, 5 mM Tris, 0.5% TFA and 0.5% MPA in 50% mobile phase (10% HPLC grade methanol, 0.09% TFA – 0.2 µm filtered). Lysates were then centrifuged for 20 min at 14,000 g, 4°C. Each sample was run in duplicate on a Pursuit5 C18 column (150 × 4.6 mm, 5 µm; Agilent Technologies, Santa Clara, CA) with a 1 mL/min flow rate and detected at 215 nm. Standards were diluted in the same buffer and interpolated between the samples. All LC/MS data were analyzed using the Masshunter software (Agilent Technologies, RRID:SCR_015040).

Media metabolite levels were determined using a BioProfile 400 Analyzer (BioNova). Briefly, 2 mL media was collected from cells after 72 hr incubation at 37°C in a $CO_2$ incubator. These were centrifuged to remove any cell debris, and 1 mL was used to measure glucose, lactate, glutamine, glutamate, $NH_4^+$, and $H^+$ levels. To control for natural degradation of metabolites, values were compared to that of media incubated in parallel wells which contained no cells.

## Integrated metabolic network analysis

Integrated metabolic network analysis was performed as previously described (*Vincent et al., 2015*) using the Shiny GAM application (https://artyomovlab.wustl.edu/shiny/gam/; *Sergushichev et al., 2016*) and visualized using Cytoscape (RRID:SCR_003032, *Shannon et al., 2003*). FDR was set to −0.25 for metabolites and −3.9 for gene expression for comparison of DoxR and Control, and to −0.1 and −3.4 respectively for comparison of EpiR and Control. Absent metabolite score was set to −0.5 for all analyses.

## ROS measurements

Cells were seeded in a 96-well dish for 48 hr prior to the experiment to achieve 75–80% confluence. Cells were maintained under normal drug conditions throughout. After PBS wash, cells were incubated with 20 µM CM-$H_2$DCFDA (Thermo Fisher Scientific #C6827) in serum-free high-glucose DMEM for 30 min at 37°C, covered in foil to prevent light exposure. Control wells without CM-$H_2$DCFDA were supplemented with equivalent volume of DMSO. After 30 min, cells were washed with PBS and incubated an additional 30 min with high-glucose DMEM supplemented with either water or 0.03% (vol/vol) $H_2O_2$. Fluorescence was then measured in an Omega plate reader (BMG LabTech) at excitation/emission wavelengths of 495/520 nm.

## Immunoblots

Total proteins from cultured cells were extracted with lysis buffer (50 mM Tris–HCl pH 7.4, 1% Triton X-100, 0.25% sodium deoxycholate, 150 mM NaCl, 1 mM EDTA) supplemented with inhibitors (2

µg/mL pepstatin, 1 µg/mL aprotinin, 1 µg/mL leupeptin, 0.2 mM phenylmethylsulfonyl fluoride and 1 mM sodium orthovanadate) and quantified using a BCA protein assay kit (Thermo Fisher Scientific #PI123225). The blots were incubated according to the manufacturer's instructions with the following primary antibodies: PGC-1α (Calbiochem #ST1202, RRID:AB_2237237), PGC-1β (Millipore #ABC218, RRID:AB_2891214), and Actin (Santa Cruz Biotechnology #sc-1616, RRID:AB_630836) and with horseradish peroxidase-conjugated secondary antibodies (anti-mouse, KPL #KP-074–1806; anti-rabbit, KPL #KP-074–1506; anti-goat, Abcam #ab6881, RRID:AB_955236). The results were visualized using Clarity ECL (Bio-Rad #1705060).

## Respirometry, bioenergetics, and $J_{ATP}$ calculations

Oxygen consumption rate (OCR) and extracellular acidification rate (ECAR) were measured using a Seahorse XFe96 Analyzer (Agilent Technologies, RRID:SCR_019545). Briefly, 10,000 cells were plated in 100 µL of their standard growth media and, after overnight culture, washed twice with XF media at pH 7.4, and equilibrated in XF media supplemented with 25 mM glucose, 4 mM glutamine, and sodium pyruvate (1X) at pH 7.4 in a $CO_2$-free 37°C incubator for 1 hr. Three sequential measurements of OCR and ECAR were taken to assay bioenergetics under basal, proton leak (1 µM oligomycin, Sigma Aldrich #O4876), maximal respiration (1 µM FCCP, Sigma Aldrich #C2920), OXPHOS inhibition (1 µM each rotenone and myxothiazol, Sigma Aldrich #R8875 and #T5580), and high glycolytic ATP demand (20 µM monensin, Sigma Aldrich #M5273) conditions. ECAR data was corrected for media buffering power as previously described (*Mookerjee et al., 2016*) and both OCR and ECAR were normalized on protein levels. ATP production rates ($J_{ATP}$), glycolytic index, bioenergetic capacity, and ATP supply flexibility were determined quantitatively as previously described (*Mookerjee et al., 2017*). Glutamine deprivations were performed over 4 hr in supplemented XF media as described above compared to media without supplemented glutamine, prior to measurement of OCR and ECAR as described.

## Immunofluorescence and quantification of mitochondrial volume

Cells were seeded onto 18 mm #1.5 glass coverslips and placed in 12-well plates overnight, then fixed with 4% PFA for 15 min at 37°C. Blocking and permeabilization was carried out by incubation with PBS containing 1% BSA and 0.5% Triton X-100. Mitochondria were visualized through staining with rabbit polyclonal anti-Tom20 antibody (Proteintech #11802–1-AP, RRID:AB_2207530) and goat anti-rabbit secondary antibody conjugated to Alexa Fluor 568 (Thermo Fisher Scientific #A-11011, RRID:AB_143157). Cytoplasm was visualized using HCS CellMask Green stain (Thermo Fisher Scientific #H32714), and nuclei were stained with DAPI. Coverslips were mounted onto microscope slides using ProLong Glass Antifade Mountant (Thermo Fisher Scientific #P36982) and kept at 4°C in the dark until imaging. Images were taken with an Axio Observer Z1 epifluorescent microscope (Zeiss), using a 63x Plan-Apochromat oil objective. Deconvolution of images was carried out in Autoquant X2 software (MediaCybernetics, RRID:SCR_002465) using an adaptive PSF with 10 iterations. Segmentation and surface rendering of mitochondria, cytoplasm, and nuclei was performed in Imaris v8 (Bitplane, RRID:SCR_007370).

## ChIP

For ChIP analyses, chromatin was prepared from Control, DoxR, and EpiR cells maintained in drug prior harvesting. Standard ChIP was performed as described previously (*Deblois et al., 2016*). Quantification of ChIP enrichment by real-time quantitative PCR was carried out using the LightCycler480 instrument (Roche). ChIPs are normalized against background enrichment on anti-IgG antibody ChIP control and average enrichment on two negative control unbound regions. The antibodies used are: anti-PGC1α (Santa Cruz Biotechnology #sc-13067, RRID:AB_2166218), anti-ERRα (Abcam #Ab76228, RRID:AB_1523580). The ChIP primers are listed in *Supplementary file 3*.

## siRNA knockdowns

Cells were subjected to either 40 nM control siRNA (Dharmacon #D-001810–10- 05) or a combined 40 nM pool of four siRNA specifically targeting *PPARGC1A* (Qiagen FlexiTube-GeneSolution #GS10891) and four siRNA specifically targeting *PPARGC1B* (Qiagen FlexiTube-GeneSolution

#GS133522). Cells were transfected using Lipofectamine RNAiMax (ThermoFisher #13778–150) and incubated for 72 hr before pursuing subsequent experiments.

## Statistical analyses

All statistical analyses were performed using GraphPad Prism (GraphPad Software Inc, RRID:SCR_002798), Microsoft Excel (Microsoft Corporation, RRID:SCR_016137), or R (R Foundation for Statistical Computing, RRID:SCR_001905 *R Development Core Team, 2019*).

## Acknowledgements

SM was recipient of a Vanier Canada Graduate Scholarship (Canadian Institutes of Health Research, CIHR), Doctoral Training Award (*Fonds de Recherche du Québec – Santé*, FRQS), Canderel Studentship Award (Goodman Cancer Research Centre, GCRC). YAD was supported by a Postdoctoral Training Award from FRQS. YX was supported by Rolande and Marcel Gosselin Graduate Studentship and Charlotte and Leo Karassik Foundation Oncology Fellowship. KZ was supported by a Doctoral Training Award (FRQS) and Gerald Clavet award (Faculty of Medicine, McGill University). JSP received salary support from FRQS and Canada Research Chair in Cancer and Metabolism. This work was supported by grants from CIHR (MOP-106603 to JSP; PJT-148650 to PS and JSP; MOP-130540 to SH) and Terry Fox Research Institute and Québec Breast Cancer Foundation (#242122 to JSP, PS, and VG). We acknowledge contributions from the Metabolomics Core Facility (MCF) of the GCRC, as well as technical assistance from Daina Avizonis, Mariana De Sa Tavares Russo, Gaëlle Bridon, and Luc Choinière. The MCF is funded by the John R and Clara M Fraser Memorial Trust, Terry Fox Research Institute and Québec Breast Cancer Foundation (#242122 to JSP, PS, and VG), and McGill University. The authors thank Amadeo Parissenti for providing resistant cell lines, and Simon-Pierre Gravel, Ouafa Najyb, David Papadopoli, Sylvia Andrzejewski, Valérie Chénard, Tina Gruosso, Uri David Akavia, Russell G Jones, and Nicole Beauchemin for thoughtful discussions. The authors extend special thanks to the staff, students, and fellows of McGill University and of the University of Ottawa who enabled a safe environment to complete this study during the COVID-19 pandemic.

## Additional information

### Funding

| Funder | Grant reference number | Author |
|---|---|---|
| Canadian Institutes of Health Research | Vanier Scholarship | Shawn McGuirk |
| Fonds de Recherche du Québec - Santé | Doctoral Training Award | Shawn McGuirk |
| McGill University | Canderel Studentship Award | Shawn McGuirk |
| Fonds de Recherche du Québec - Santé | Postdoctoral Training Award | Yannick Audet-Delage |
| McGill University | Charlotte & Leo Karassik Foundation Oncology Fellowship | Yibo Xue |
| McGill University | Rolande & Marcel Gosselin Graduate Studentship | Yibo Xue |
| Fonds de Recherche du Québec - Santé | Doctoral Training Award | Kaiqiong Zhao |
| McGill University | Gerald Clavet Award | Kaiqiong Zhao |
| Fonds de Recherche du Québec - Santé | Salary Award | Julie St-Pierre |
| Canada Research Chairs | Tier 1 - Cancer and Metabolism | Julie St-Pierre |
| Canadian Institutes of Health | MOP-106603 | Julie St-Pierre |

| | | |
|---|---|---|
| Research | | |
| Canadian Institutes of Health Research | PJT-148650 | Peter M Siegel<br>Julie St-Pierre |
| Canadian Institutes of Health Research | MOP-130540 | Sidong Huang |
| Terry Fox Research Institute | 242122 | Vincent Giguère<br>Peter M Siegel<br>Julie St-Pierre |
| Quebec Breast Cancer Foundation | Grant with TFRI | Vincent Giguère<br>Peter M Siegel<br>Julie St-Pierre |

The funders had no role in study design, data collection and interpretation, or the decision to submit the work for publication.

### Author contributions
Shawn McGuirk, Conceptualization, Resources, Data curation, Formal analysis, Funding acquisition, Validation, Investigation, Visualization, Methodology, Writing - original draft, Project administration, Writing - review and editing; Yannick Audet-Delage, Formal analysis, Validation, Investigation, Visualization, Methodology, Writing - original draft, Project administration, Writing - review and editing; Matthew G Annis, Investigation, Methodology, Writing - review and editing; Yibo Xue, Investigation, Methodology; Mathieu Vernier, Formal analysis, Investigation, Visualization, Methodology, Writing - review and editing; Kaiqiong Zhao, Formal analysis, Visualization; Catherine St-Louis, Lucía Minarrieta, Investigation, Writing - review and editing; David A Patten, Formal analysis, Validation, Writing - review and editing; Geneviève Morin, Methodology; Celia MT Greenwood, Resources, Supervision, Methodology, Writing - review and editing; Vincent Giguère, Peter M Siegel, Resources, Supervision, Funding acquisition, Writing - review and editing; Sidong Huang, Resources, Formal analysis, Supervision, Funding acquisition, Visualization, Methodology, Writing - review and editing; Julie St-Pierre, Conceptualization, Resources, Supervision, Funding acquisition, Visualization, Writing - original draft, Project administration, Writing - review and editing

### Author ORCIDs
Shawn McGuirk (iD) https://orcid.org/0000-0002-7183-7962
Yannick Audet-Delage (iD) http://orcid.org/0000-0002-8467-6168
Matthew G Annis (iD) http://orcid.org/0000-0002-8776-004X
Yibo Xue (iD) http://orcid.org/0000-0003-4252-4446
Mathieu Vernier (iD) http://orcid.org/0000-0001-9356-7353
Celia MT Greenwood (iD) https://orcid.org/0000-0002-2427-5696
Sidong Huang (iD) http://orcid.org/0000-0002-2838-4726
Peter M Siegel (iD) http://orcid.org/0000-0002-5568-6586
Julie St-Pierre (iD) https://orcid.org/0000-0002-2815-7099

### Ethics
Animal experimentation: Mice were housed in facilities managed by the McGill University Animal Resources Centre and all animal experiments were conducted under a University approved animal use protocol (AUP2012-5129) in accordance with guidelines established by the Canadian Council on Animal Care.

### Decision letter and Author response
Decision letter https://doi.org/10.7554/eLife.65150.sa1
Author response https://doi.org/10.7554/eLife.65150.sa2

# Additional files

## Supplementary files

- Supplementary file 1. PGC-1α supports therapeutic resistance across several cancer types.
- Supplementary file 2. List of primer sequences for RT-qPCR.
- Supplementary file 3. List of primer sequences for ChIP.
- Transparent reporting form

## Data availability

Microarray data have been deposited in GEO under accession code GSE125187.

The following dataset was generated:

| Author(s) | Year | Dataset title | Dataset URL | Database and Identifier |
|---|---|---|---|---|
| McGuirk S, St-Pierre J | 2019 | Gene expression data in Control, Doxorubicin-resistant, and Epirubicin-resistant breast cancer cells | https://www.ncbi.nlm.nih.gov/geo/query/acc.cgi?acc=GSE125187 | NCBI Gene Expression Omnibus, GSE125187 |

The following previously published dataset was used:

| Author(s) | Year | Dataset title | Dataset URL | Database and Identifier |
|---|---|---|---|---|
| Gruosso T, Kieffer Y, Mechta-Grigoriou F | 2016 | Response to Neoadjuvant Chemotherapy in Triple Negative Breast tumors | https://www.ncbi.nlm.nih.gov/geo/query/acc.cgi?acc=GSE43816 | NCBI Gene Expression Omnibus, GSE43816 |

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

# Appendix 1

**Appendix 1—key resources table**

| Reagent type (species) or resource | Designation | Source or reference | Identifiers | Additional information |
|---|---|---|---|---|
| Cell line (*H. sapiens*) | MCF-7 Control, MCF-7$_{CC}$ | Obtained from Dr. Amadeo Parissenti ***Hembruff et al., 2008***; DOI: 10.1186/1471-2407-8-318 | | Grown in 0.1% DMSO media. Negative for mycoplasma. |
| Cell line (*H. sapiens*) | MCF-7 DoxR, MCF-7$_{DOX-2}$ | Obtained from Dr. Amadeo Parissenti ***Hembruff et al., 2008***; DOI: 10.1186/1471-2407-8-318 | | Resistant to 98.1 nM doxorubicin. Negative for mycoplasma. |
| Cell line (*H. sapiens*) | MCF-7 EpiR, MCF-7$_{EPI}$ | Obtained from Dr. Amadeo Parissenti ***Hembruff et al., 2008***; DOI: 10.1186/1471-2407-8-318 | | Resistant to 852 nM epirubicin. Negative for mycoplasma. |
| Cell line (*H. sapiens*) | MCF-7 Ctl | This paper | | Grown in 0.1% DMSO media. Derived from MCF-7 cells obtained from the American Type Culture Collection (ATCC). Negative for mycoplasma. |
| Cell line (*H. sapiens*) | MCF-7 D100 | This paper | | Resistant to 100 nM doxorubicin. Derived from MCF-7 cells obtained from the American Type Culture Collection (ATCC). Negative for mycoplasma. |
| Cell line (*H. sapiens*) | MCF-7 E100 | This paper | | Resistant to 100 nM epirubicin. Derived from MCF-7 cells obtained from the American Type Culture Collection (ATCC). Negative for mycoplasma. |
| Antibody | Human PGC-1α (mouse, monoclonal) | Calbiochem | Cat #: ST1202; RRID:AB_2237237 | Immunoblots, (1:1000) |
| Antibody | Human PGC-1β (rabbit, polyclonal) | Millipore | Cat #: ABC218 RRID:AB_2891214 | Immunoblots, (1:1000) |
| Antibody | Human Actin (goat, polyclonal) | Santa Cruz Biotechnology | Cat #: sc-1616 RRID:AB_630836 | Immunoblots, (1:2000) |
| Antibody | anti-mouse (goat, polyclonal) | KPL | Cat #: KP-074-1806 | Immunoblots, (1:10000) |
| Antibody | anti-rabbit (goat, polyclonal) | KPL | Cat #:KP-074-1506 | Immunoblots, (1:10000) |
| Antibody | anti-goat (donkey, polyclonal) | Abcam | Cat #: ab6881 RRID:AB_955236 | Immunoblots, (1:10000) |
| Antibody | Human Tom20 (rabbit, polyclonal) | Proteintech | Cat #:11802-1-AP RRID:AB_2207530 | Immuno-fluorescence |

*Continued on next page*

*Appendix 1—key resources table continued*

| Reagent type (species) or resource | Designation | Source or reference | Identifiers | Additional information |
|---|---|---|---|---|
| Antibody | anti-rabbit conjugated to Alexa Fluor 568 (goat, polyclonal) | Thermo Fisher Scientific | Cat #: A-11011 RRID:AB_143157 | Immuno-fluorescence |
| Antibody | Human PGC-1α (rabbit, polyclonal) | Santa Cruz Biotechnology | Cat #: sc-13067 RRID:AB_2166218 | ChIP |
| Antibody | Human ERRα (rabbit, monoclonal) | Abcam | Cat #: Ab76228 RRID:AB_1523580 | ChIP |
| transfected construct (*H. sapiens*) | ON-TARGETplus Non-targeting Control Pool siRNA | Dharmacon | Cat #: D-001810-10-05 | 40 nM pool of siRNA |
| transfected construct (*H. sapiens*) | ON-TARGETplus Human PPARGC1A siRNA | Qiagen | Cat #: FlexiTube-Gene SolutionGS10891 | Combined 40 nM pool of four siRNA |
| transfected construct (*H. sapiens*) | ON-TARGETplus Human PPARGC1B siRNA | Qiagen | Cat #: FlexiTube-Gene SolutionGS133522 | Combined 40 nM pool of four siRNA |
| transfected construct (*H. sapiens*) | FDA shRNA library | The McGill Platform for Cell Perturbation (MPCP) of the Rosalind and Morris Goodman Cancer Research Centre and Biochemistry department at McGill University | | Developed by YX, GM, and SH |
| sequence-based reagent | RT-qPCR primers | | | See *Supplementary file 2* |
| sequence-based reagent | ChIP primers | | | See *Supplementary file 3* |
| commercial assay or kit | Aurum Total RNA Mini Kit | Bio-Rad | | |
| commercial assay or kit | iScript cDNA Synthesis kit | Bio-Rad | | |
| commercial assay or kit | iQ SYBR Green Supermix | Bio-Rad | | |
| commercial assay or kit | BCA protein assay kit | Thermo Fisher Scientific | Cat #: PI123225 | |
| commercial assay or kit | Seahorse XFe96 Analyzer | Agilent Technologies | RRID:SCR_019545 | |
| commercial assay or kit | BioProfile 400 Analyzer | BioNova | | |
| chemical compound, drug | Doxorubicin | AbMole Biosciences | Cat #: M1969 | |
| chemical compound, drug | Epirubicin | Sigma Aldrich | Cat #: E9406 | |

*Continued on next page*

*Appendix 1—key resources table continued*

| Reagent type (species) or resource | Designation | Source or reference | Identifiers | Additional information |
|---|---|---|---|---|
| chemical compound, drug | L-buthionine-sulfoximine | Sigma Aldrich | Cat #: B2515 | |
| chemical compound, drug | Phenformin | Sigma Aldrich | Cat #: P7045 | |
| chemical compound, drug | Oligomycin | Sigma Aldrich | Cat #: O4876 | |
| chemical compound, drug | FCCP (Carbonyl cyanide 4-(trifluoromethoxy) phenylhydrazone) | Sigma Aldrich | Cat #: C2920 | |
| chemical compound, drug | Rotenone | Sigma Aldrich | Cat #: R8875 | |
| chemical compound, drug | Myxothiazol | Sigma Aldrich | Cat #: T5580 | |
| chemical compound, drug | Monensin | Sigma Aldrich | Cat #: M5273 | |
| chemical compound, drug | CM-H$_2$DCFDA | Thermo Fisher Scientific | Cat #: C6827 | |
| chemical compound, drug | Clarity ECL | Bio-Rad | Cat #: 1705060 | |
| chemical compound, drug | HCS CellMask Green stain | Thermo Fisher Scientific | Cat #: H32714 | |
| chemical compound, drug | Lipofectamine RNAiMax | Thermo Fisher Scientific | Cat #: 13778-50 | |
| chemical compound, drug | ProLong Glass Antifade Mountant | Thermo Fisher Scientific | Cat #: P36982 | |
| software, algorithm | Autoquant X2 software | MediaCybernetics | RRID:SCR_002465 | |
| software, algorithm | Imaris v8 | Bitplane | RRID:SCR_007370 | |
| software, algorithm | GraphPad Prism | GraphPad Software, Inc | RRID:SCR_002798 | |
| software, algorithm | Microsoft Excel | Microsoft Corporation | RRID:SCR_016137 | |
| software, algorithm | R Project for Statistical Computing | R Foundation for Statistical Computing, (*R Development Core Team, 2019*) | RRID:SCR_001905 | |
| software, algorithm | R Bioconductor | DOI:10.1186/gb-2004-5-10-r80 | RRID:SCR_006442 | |

*Continued on next page*

*Appendix 1—key resources table continued*

| Reagent type (species) or resource | Designation | Source or reference | Identifiers | Additional information |
|---|---|---|---|---|
| software, algorithm | Shiny GAM | https://artyomovlab.wustl.edu/shiny/gam/ *Sergushichev et al., 2016*; DOI:10.1093/nar/gkw266 | | |
| software, algorithm | Cytoscape | *Shannon et al., 2003*; DOI:10.1101/gr.1239303 | RRID:SCR_003032 | |
| software, algorithm | In-house algorithm for isotopic corrections | In-house algorithm of the St-Pierre laboratory, first described in *McGuirk et al., 2013*; DOI:10.1186/2049-3002-1-22 | | Developed by SM |
| software, algorithm | Masshunter Quantitative Analysis software | Agilent Technologies | RRID:SCR_015040 | |
| software, algorithm | Chemstation software | Agilent Technologies | RRID:SCR_015742 | |
| software, algorithm | Transcriptome Analysis Console | Affymetrix | RRID:SCR_016519 | |
| software, algorithm | Gene Set Enrichment Analysis | *Mootha et al., 2003*; *Subramanian et al., 2005*; DOI:10.1038/ng1180, 10.1073/pnas.0506580102 | RRID:SCR_003199 | |
| software, algorithm | Gene Expression Omnibus | NCBI *Edgar et al., 2002*; DOI:10.1093/nar/30.1.207 | RRID:SCR_005012 | |
| software, algorithm | MAGeCK (v0.5.5) | *Li et al., 2014*; DOI:10.1186/s13059-014-0554-4 | | |
| other | GEO patient dataset | GSE43816 *Gruosso et al., 2016*; DOI:10.15252/emmm.201505891 | | Gene expression of human breast cancer tumors biopsies prior to and after treatment with four cycles of epirubicin and cyclophosphamide, followed by four cycles of docetaxel |
| other | GEO cell line dataset | GSE125187; this paper | | Gene expression of Control, DoxR, and EpiR cells |

