## [Decision Letter]

**Acceptance summary:**

In this paper, the authors show that doxorubicin and epirubicin resistant cells have different metabolic characteristics. Specifically, they show that doxorubicin resistant cells rely on increased glutamine consumption to produce mitochondrial ATP and to synthesize glutathione whereas epirubicin resistant cells do not. The authors speculate that the unique metabolic characteristics associated with doxorubicin resistance are caused by the induction of more oxidative stress, but exactly what drives this difference will be a topic for future work.

**Decision letter after peer review:**

[Editors’ note: the authors submitted for reconsideration following the decision after peer review. What follows is the decision letter after the first round of review.]

Thank you for submitting your work entitled "Tailored metabolic adaptations confer resistance to chemotherapy in breast cancer" for consideration by *eLife*. Your article has been reviewed by 3 peer reviewers, one of whom is a member of our Board of Reviewing Editors, and the evaluation has been overseen by a Senior Editor. The reviewers have opted to remain anonymous.

Our decision has been reached after consultation between the reviewers. Based on these discussions and the individual reviews below, we regret to inform you that your work will not be considered further for publication in *eLife*. While all reviewers found the topic to be of interest, our concern is that the time required to further justify the conclusions was such that you may wish to seek publication elsewhere. If you feel that revisions could be done to address the concerns of the reviewers, then such a resubmission will be overseen by the same group of reviewers and editors (if possible). Please note that we cannot guarantee the outcome of a resubmission as a new manuscript, but minimally, the following points would have to be addressed:

1. Titration of drug dose to address whether the metabolic adaptation is related to the dose or the drug itself.

2. Further work to establish causality for the specific metabolic alterations in resistance to doxorubicin versus epirubicin.

3. Some additional effort to assess the relevance of these resistance mechanisms in vivo. The reviewers felt that the patient correlation included in the study is contrary to the claim that specific metabolic adaptations arise to each drug, and thus some work to show the main claims hold in some in vivo context is needed.

*Reviewer #1:*

The authors select for doxorubicin and epirubicin resistant MCF7 cells in culture and determine that the global gene expression profile of these cells is similar to tumors from patients exposed to anthracylins. They then characterize the resistant cells, and argue that doxorubicin or epirubicin select for different adaptations, which is surprising given the similarity in these drugs.

1. Is the difference between doxorubicin response and epirubicin response simply drug dose? The bulk of the analysis largely compares a cells from a single dose where doxorubicin is cytostatic and epirubicin is cytotoxic. If doses of either drug are titrated to show cytostasis or cytotoxicity is there still a difference in metabolic adaptation of the resistant cells? This is only partially addressed in Figure 7, and is a crucial questions that is important to judge the central claim of the paper.

2. There is a large literature arguing that response to chemotherapy in culture is not necessarily indicative of response in patients, and there is a growing literature around cancer metabolism also being different in the cell culture and in tumors, including glutamine metabolism. The bulk of the analysis is based on cell culture, and if some data were available that speak to whether these same differences or dependencies hold in vivo it would greatly improve the manuscript.

3. The authors assess labeling of metabolites at short time points (15 min), which can be more useful for assessment of flux through pathways like glycolysis. However, this data would be much more informative if there were multiple time points assessed, as this is typically need to interpret kinetic tracing data.

4. Seahorse assays were used to suggest ATP production rate, but this approach requires several assumptions that should be more explicitly discussed. I suggest showing the raw data, with or without the calculated rates, and feel this is particularly important for a broad audience who may not appreciate the caveats associated with this analysis.

5. While the fact that targeting PGC1a overcomes resistance to both doxorubicin and epirubicin is potentially relevant for clinical medicine, from a mechanistic standpoint it does not support their claim that these drugs elicit different metabolic adaptations. Is there an example of a target that separates the contexts as evidence of differential resistance mechanism?

*Reviewer #2:*

In their manuscript, "Tailored Metabolic Adaptations Confer Resistance to Chemotherapy in Breast Cancer", McGuirk et al. derive breast cancer cells resistant to doxorubicin and epirubicin and investigate whether the metabolic alterations that occur during resistance are dependent on the therapeutic. They suggest that doxorubicin resistant cells rely on glutamine for mitochondrial function and glutathione synthesis, while epirubicin resistant cells increase their mitochondrial capacity and ATP production. They suggest that targeting PGC1a could eradicate cells resistant to both therapeutics, as PGC1a regulates both metabolic adaptions. While the idea that different therapeutics of the same class could cause resistance due to different metabolic mechanisms, this study was limited by a lack of evidence for a causal role for the metabolic alterations identified in drug resistance.

1. It was difficult to follow the claims of the manuscript. Many statements are vague and often the figure legends and section headings/text do not match, and are sometimes contradictory in their claims.

2. The authors generate doxorubicin resistant (DoxR) and epirubicin resistant (EpiR) resistant cell lines via exposure to increasing concentrations of drug. DoxR resistant cells are cultured in a final concentration of 98.1nM drug, while epirubicin resistant cells are cultured in 852nM drug. Treatment of chemonaive cells with these concentrations results in cytostatic (Dox) or cytotoxic (Epi) effects. This approach raises several concerns.

a. First, why these concentrations? These differing concentrations are a concern for the major findings of the manuscript.

b. Second, anthracyclins have been shown to damage mitochondrial DNA, and the use of differing concentrations may result in different amounts of damage. Related to this point does compensatory mitochondrial biogenesis explain why PGC1a expression is increased, particularly in the EpiR cells, which are treated with significantly more anthracycline? What happens to mitochondrial content in these resistant cell lines? Mitochondrially encoded respiratory components? How does PGC1a/b knockdown affect mitochondrial content?

c. Third, the final figure, in which new resistant cell lines are derived and grown in the same amount of Dox or Epi, the authors show now that while DoxR cells still increase GSH synthesis from glucose, similar increases in mitochondrial ATP and bioenergetic capacity are observed between DoxR and EpiR cells. While the effects are modestly more pronounced in EpiR cells, this finding is contrary to the thesis that Dox and Epi resistance induce different metabolic alterations as it relates to the mitochondria. These cell lines should be used throughout the study to eliminate any confounding effects of differing drug concentrations.

3. Related to the point above, it is unclear how PGC1a is promoting resistance to doxorubicin and epirubicin. In Figure 6, PGC1a associates with the antioxidant response genes only in EpiR cells, but PGC1a/b knockdown has a greater effect on gene expression in the DoxR cells. What is the effect of PGC1a knockdown alone? In addition, Figure 3i shows that EpiR cells have little to no elevation of antioxidant gene expression, despite significantly increased PGC1a binding to their promoters. How are these findings reconciled? Further, siPGC1a/b affects the growth of DoxR and EpiR cells equally. Is this in the presence of drug? It is unclear from the figure legend. Finally, is PGC1a overexpression sufficient to promote resistance of drug naïve cells? This experiment is needed to show causation.

4. The authors suggest that cells with engineered resistance to Dox and Epi have similar gene expression patterns to residual tumors of breast cancer patients following anthracycline-based chemotherapy, but there are several problems with this analysis/these claims. First, it is not clear that these tumors are resistant to anthracyclines. From the description, patients received 4 cycles of anthracycline-based therapy, followed by four cycles of docetaxel. Resultant tumors may be responders, resistant to anthracyclines, resistant to docetaxel, or both. As a consequence, it is difficult to draw meaningful conclusions from the comparison. Second, the authors compare their cell line gene expression to these tumors, but tumors were treated only with epirubicin. Since the authors argue that resistance to Dox vs. Epi is via discrete mechanisms, the argument for similarity between DoxR cells and Epi "resistant" tumors is contrary to the main claims of the manuscript. Indeed, the claim that "several distinct metabolic pathways were altered by doxorubicin or epirubicin resistance (Figure 1f,h)" is problematic due to these concerns since (1) these pathways are based on overlap with gene expression from tumors from patients that may or may not be resistant and (2) these tumors were treated only with epirubicin. Therefore, the basis for distinct metabolic pathways is unsound.

5. The increase in GCLC, ME1, NQO1 etc in DoxR cells is consistent with NRF2 activation as proposed, as is the increase in GSH M+5, which suggests an increase in GSH synthesis. However, the decrease in the GSH/GSSG ratio and NADPH, and increase in NADP+ is not consistent, nor is it consistent with the protection against ROS shown in Figures 3j and 3k. How is this reconciled? What is the total pool size of GSH + GSSG? Related to this, what is the "control" for figures 3J and 3k? Parental cells?

6. The data shown in Figure 4 is not supportive of an increased dependence of DoxR cells on glutamine, as claimed in the figure legend. Glutamine is used for many purposes in cells and neither DoxR nor EpiR cells grow without glutamine in Figure 4A. Further, the idea that glutamine is driving TCA cycle is not matched by the tracing data from Figure 3. There is no increased entry of glucose or glutamine into the TCA cycle in the DoxR cells (e.g. citrate m+2 from glucose, citrate m+4 from glutamine). Related to this, the inability of DoxR cells to compensate for ATP production upon glutamine withdrawal suggests a defect in glycolytic regulation or ATP sensing, rather than glutamine metabolism. More clarity is needed for (1) what metabolic alterations occur, (2) how they are regulated and (3) how they promote anthracycline resistance.

*Reviewer #3:*

In "Tailored Metabolic Adaptations Confer Resistance to Chemotherapy in Breast Cancer", McGuirk et al. use MCF-7 cells in culture to study mechanisms of resistance to doxorubicin and epirubicin. Overall, I found their approach to be intriguing. It seems like a compelling approach to study drug resistance in cancer. They subjected cells to many passages in the presence of drug, which resulted in a population of resistant cells. They could then compare these resulting resistant cells to the original drug-susceptible cells. In doing so, the authors claim to have found different metabolic adaptations to each drug. Given the differences in adaptation, they suggest that it might be better to target resistant tumors with drugs that hit global regulators of metabolism, like PGC-1a.

I have two major questions.

1. How do the authors know that the genes and metabolic adaptations assumed in the resistant cells are essential to drug resistance? Might it be the case that their original control population of cells was heterogeneous and exposure of cells to drug over long periods of time simply selected those subclones that were resistant? In such a model, while one gene/metabolic phenotype may be essential to developing resistance (perhaps ABC transporters), the other genes/metabolic phenotypes could be random. Indeed, based on the limited amount of data shown from two experiments, it seems like the metabolic phenotypes of cells exposed to dox for long periods of time are different in separate experiments. For example, the isotope labeling shown in Figure 3 looks very different from the labeling shown in Figure 7. A recent study by Speirs and Price et al. (doi: 10.18632/oncotarget.26533) did a similar experiment without drugs. After selecting subclones of cells from a culture, they found significant differences in metabolic phenotypes. It would be most interesting to compare the genetic and metabolic differences of resistance cells that were selected independently by drugs in separate experiments. This could help better resolve whether the differences observed are essential.

2. The major premise of this manuscript is that the authors have selected cells resistant to dox and epi. Data shown in Figure 1B suggest that this is the case over 6 days. Based on the data shown in Figure 4A, however, I worry that the Epi-R cells are not resistant to Epi over long periods of time. For example, at day 11, the Epi-R live cell count looks more like the control cells in Figure 1 than DoxR. Thus, if the cells are not both equally resistant, maybe the metabolic adaptations are not really "distinct" but rather just reflective of different degrees of resistance.

Other points:

1. Can the authors do any experiments to explore (or possibly speculate in the discussion) why the two drugs would elicit different resistance mechanisms?

2. The increase of pyruvate carboxylase in EpiR cells is interesting. What is the phenotypic value of such a metabolic alteration in resistance?

3. The data do not seem to support this statement: "DoxR cells were unable to increase glycolytic ATP production to compensate for diminished oxidative ATP production upon glutamine withdrawal." The Seahorse plots in Figure 4 actually show that glycolytic ATP production increases in drug resistance.

4. It would be helpful to see all of the isotopologues, instead of just a select number. For example, M+3 in malate from 13C-glucose doesn't necessarily mean pyruvate carboxylase activity. It could also be indicative of two rounds of TCA cycle (the first with labeled acetyl-CoA and the second with unlabeled). It would be easier to assess these kinds of possibilities if the authors presented full isotopologue plots.

5. Many of the interesting data shown are not explored (or discussed). For example, in Figure 5C-D, the DoxR cells have uncoupled mitochondria. Why is this? Is this the source of ROS? It might be best to remove data that are not discussed, as it is distracting from the overall message.

[Editors’ note: further revisions were suggested prior to acceptance, as described below.]

Thank you for submitting your article "Resistance to different anthracyclines elicits distinct and actionable primary metabolic dependencies in breast cancer" for consideration by *eLife*. Your article has been reviewed by 3 peer reviewers, one of whom is a member of our Board of Reviewing Editors, and the evaluation has been overseen by Utpal Banerjee as the Senior Editor. The reviewers have opted to remain anonymous.

The reviewers have discussed the reviews with one another and the Reviewing Editor has drafted this decision to help you prepare a revised submission.

As the editors have judged that your manuscript is of interest, and much improved from the prior version, the reviewers felt that additional experiments are required before it is published, including making more extensive use of the independently derived resistant cells. We would like to draw your attention to changes in our revision policy that we have made in response to COVID-19 (https://elifesciences.org/articles/57162). First, because many researchers have temporarily lost access to the labs, we will give authors as much time as they need to submit revised manuscripts. We are also offering, if you choose, to post the manuscript to bioRxiv (if it is not already there) along with this decision letter and a formal designation that the manuscript is "in revision at *eLife*". Please let us know if you would like to pursue this option.

Summary:

In this paper, the authors show that doxorubicin and epirubicin resistant cells have different metabolic characteristics. Specifically, they show that doxorubicin resistant cells rely on increased glutamine consumption to produce mitochondrial ATP and to synthesize glutathione whereas epirubicin resistant cells do not. The authors speculate that the unique metabolic characteristics associated with doxorubicin resistance are caused by the induction of more oxidative stress, but exactly what drives this difference remains unknown.

Essential revisions:

1. Concern remains regarding how relevant the metabolic differences between these two cell populations are to resistance in patients given that different drug concentrations were used to derive the EpiR and DoxR cells and that the new in vivo data did not use a de novo resistance model, but rather the same cell lines used previously. The use of independently derived cell lines should be used to confirm more of the central findings of the manuscript. In particular, please address whether the different metabolic adaptations are related to the dose or the drug itself.

2. The tumor growth differences are not impressive when considering that the DoxR cells grow much more slowly than the EpiR cells. The difference in size even for the DoxR tumors does not seem that great, and is presented as fold change. Please at least acknowledge this, consider presenting the data in a more fair way with tumor size data graphed as actual measured size with the DoxR and EpiR curves on the same plot or plots with the same scales.

3. Please test (in vitro) whether the oxidative stress response in control cells treated with either doxorubicin or epirubicin is different as a potential mechanism for how treatment with these drugs differs in a way that might impact metabolism. Please also considering examining whether Epic cells are more sensitive to OXPHOS inhibitors to consider whether PGC1a promoting the OXPHOS phenotype in epi resistant cells speaks to the divergent metabolism.

4. Please address the issues with the resistance mechanisms in vivo based on the following comment from Reviewer #3. The in vivo BSO experiment is supportive of the cell culture experiment but there are multiple issues with this experiment. First, no anthracycline is included in this experiment making its relevance to cancer therapy questionable. Second, resistant lines are used from the start, rather than assaying the role of GSH synthesis in de novo resistance that arises in vivo, which would have been more relevant to the patient situation.

The full set of comments from the reviewers is provided below for your reference.

*Reviewer #1:*

The authors characterize the metabolism and gene expression of doxorubicin and epirubicin resistant MCF7 cells in culture and present data that differences exist in the cells that are resistant to these drugs from the same chemotherapy class. They show that independently derived resistant lines retains many of the same phenotypic differences. Finally, they show these differences can lead to different therapeutic vulnerabilities, including response to inhibitors of glutathione synthesis.

A strength is characterizing how cells can gain resistance to similar classes of drugs. The work focuses more on the differences than the similarities, and more analysis is needed to relate their findings to resistance that arises in patients following treatment. There is also suggestion that tumors derived from the different resistant cell lines grow very differently, and this may or may not be part of the differnces in sensitivity to inhibitors of glutathione synthesis.

1. The data relating the DoxR and EpiR gene signatures to gene expression data from patients receiving anthracyclines is use to argue human relevance of the models, although it is unclear to what extent the patient data can be divided into those who received doxorubicin or epirubicin. It seems this is relating the resistance to anthracyclines in general, which speaks more to the similarity in resistance mechanisms rather than the differences that are highlighted in the title and abstract.

2. The tumor growth differences are not impressive when considering that the DoxR cells grow much more slowly than the EpiR cells. One might question if a similar response to BSO would be observed in the EpiR cells if treatment was started at a size that is matched to the DoxR tumors. The difference in size even for the DoxR tumors does not seem that great, and is presented as fold change. At least acknowledging this is needed, and ideally to avoid misleading readers the tumor size data should be graphed as actual measured size with the DoxR and EpiR curves on the same plot (or plots with the same scales).

3. I do not think a difference in glucose to lactate flux is supported by the tracing data. Time to steady state labeling is the best measure of flux and that appears very similar between DoxR and EpiR cells. The differences are based on what is variation in only one time point. This is not central to the work, but should be presented accurately.

4. In this Reviewers opinion, terms like "increased bioenergetic capacity" are not very helpful in understanding what is biologically different. It is acknowledged they are used by many, so is not fair to ask these authors to not use them, but wanted to mention this nevertheless.

*Reviewer #2:*

The authors show that doxorubicin and epirubicin resistant cells have different metabolic characteristics. Specifically, they show that doxorubicin resistant cells rely on increased glutamine consumption to produce mitochondrial ATP and to synthesize glutathione whereas epirubicin resistant cells do not. As noted above, this is a re-submission that has been improved. The major strengths of the work are the breadth of results showing that doxorubicin and epirubicin resistant cells have different metabolic characteristics – including gene expression data, metabolomics data, isotope tracing data, and tumor data from a mouse model. The authors generally did a nice job of responding to previous comments made by reviewers and revising language to improve clarity. The weakness of the work is that the mechanism(s) underlying the observations are not well defined. The authors speculate that the unique metabolic characteristics associated with doxorubicin resistance are caused because doxorubicin's mode of action induces more oxidative stress than epirubicin, but this key assumption is not proven and only rationalized anecdotally on the basis of cardiotoxicity patient data. Before publication, I recommend an in vitro experiment to test this idea where the authors assess the oxidative stress response in control cells treated with either doxorubicin or epirubicin.

Some other comments and questions:

1. Some sentences in the abstract might be improved by making them less general.

2. In several parts of the manuscript, it is implied that glycolysis is favored in cancer cells over oxphos. This is not necessarily true for all cancer cells and tumors.

3. The authors state that EpiR cells rely on cysteine metabolism (Figure 1k). Based on their BSO data, presumably this isn't for glutathione synthesis. Why are EpiR cells sensitive to cysteine metabolism?

4. Glutathione can be oxidized/reduced to buffer oxidative stress, but it does not provide the reducing equivalents. For that, synthesis of NADPH is needed. One possibility is that glucose is re-routed to the pentose phosphate pathway in DoxR cells, thereby increasing the need to fuel mitochondrial metabolism with glutamine.

5. The authors state that IDH1 activity is increased in DoxR cells, but the evidence presented is relatively weak.

*Reviewer #3:*

In their manuscript, "Tailored Metabolic Adaptations Confer Resistance to Chemotherapy in Breast Cancer", McGuirk et al. present a revised version of their study that seeks to understand the divergent metabolic changes in doxorubicin and epirubicin resistant cells. Generally, the writing of the manuscript is much improved, the claims are much easier to follow, and the presentation of the data is more focused and digestible. Many of the confusing and problematic claims have been removed, and the description of the data is more accurate. However, the major issues with this study from the previous review remain.

1. Following their previous submission, it was suggested that the authors titrate the drug dose to address whether the metabolic adaptation is related to the dose or the drug itself. Because the dose of epi used to generate resistant lines was almost 10x greater than dox, and provoke different responses in naïve cells (cytotoxicity vs cytostasis) the resulting metabolic changes may simply be attributed to differences in drug dosing. However, this point remains unaddressed. Rather than expanding upon the independently derived resistant lines generated to the same final stable concentration of drug (100nM), the authors buried this data in the supplemental with insufficient discussion.

2. It was also suggested that the authors perform additional work to establish causality for the specific metabolic alterations in resistance to doxorubicin versus epirubicin. The authors have added BSO, an inhibitor of glutathione synthesis, which they show selectively impairs the viability of dox resistant cells, but not epi resistant cells, which supports that resistance-induced metabolic rewiring is targetable. However, the role of PGC1a in promoting the OXPHOS phenotype in epi resistant cells is not addressed; nor is it demonstrated that PGC1a is sufficient to drive the OXPHOS phenotype; nor is it examined whether epi resistant cells are more sensitive to OXPHOS inhibitors. Consequently, this aspect of the revised manuscript focuses primarily on dox resistance and not the divergent metabolic adaptations as intended.

3. Finally, it was suggested that the relevance of the resistance mechanisms in vivo required additional work. The in vivo BSO experiment is supportive of the cell culture experiment but there are multiple issues with this experiment. First, no anthracycline is included in this experiment making its relevance to cancer therapy questionable. Second, resistant lines are used from the start, rather than assaying the role of GSH synthesis in de novo resistance that arises in vivo, which would have been more relevant to the patient situation. Finally, the role of PGC1a/OXPHOS in epi resistance was not addressed in vivo.

---

## [Author Response]

[Editors’ note: the authors resubmitted a revised version of the paper for consideration. What follows is the authors’ response to the first round of review.]1. Titration of drug dose to address whether the metabolic adaptation is related to the dose or the drug itself.

We acknowledge the concerns of the reviewers and have provided clarifications regarding the cell line models. The information below supports that the metabolic adaptations reported in our manuscript are linked to the drug itself and not the drug concentration.

The DoxR and EpiR cell models were first described by the Parissenti group in a 2008 study as outlined in our Methods section (https://dx.doi.org/10.1186%2F1471-2407-8318), and have been used in several publications over the years (for example:https://dx.doi.org/10.1158%2F0008-5472.CAN-16-0774, https://dx.doi.org/10.1038%2Fs41598-018-23496-y, https://bmccancer.biomedcentral.com/articles/10.1186/s12885-016-2790-3). These models were selected over hundreds of passages over a twelve-dose process.

Doxorubicin-resistant cells (DoxR, originally named MCF-7_DOX-2_) were selected up to the highest concentration of drug at which the cells could survive (the maximally tolerated dose, 98.1 nM doxorubicin), whereas Epirubicin-resistant cells (EpiR, originally named MCF-7_EPI_) were still viable at the concentration reached at step twelve (852 nM epirubicin).

As shown by the Parissenti’s group, the initial acquisition of resistance to doxorubicin and epirubicin was linked to reduction in drug accumulation by the cells. However, at higher doses the development of resistance was not linked to drug exclusion by the cells. Indeed, Figure 2 from Hembruff et al., 2008 clearly shows that the magnitude of resistance was not linked to reduction in drug accumulation after the initial acquisition, as indicated by the plateau.

Furthermore, forcing the accumulation of doxorubicin in DoxR cells using Cyclosporin A (CsA) did not significantly impact the sensitivity of DoxR cells (solid and dotted blue lines in Figure 5A, B in Hembruff et al., 2008). These data clearly highlight the importance of adaptation mechanisms separate from drug exclusion, and that once the cells are stably resistant, increasing drug concentration has no significant impact on drug sensitivity.

These elements considered, and given the fact that DoxR cells are more resistant to doxorubicin (resistance factor, RF=27.8) than epirubicin (RF=4.79), and that EpiR cells are more resistant to epirubicin (RF=815.3) than doxorubicin (RF=203.4), we hypothesized that the mechanisms of resistance may differ for doxorubicin and epirubicin, despite the chemical similarity of the drugs.

Third and finally, we further addressed this potential impact of difference in the dose by independently deriving cells resistant to 100nM of either doxorubicin or epirubicin. With these cells, we recapitulated the main findings of DoxR and EpiR cells, upholding the conclusions of the paper, i.e. that EpiR cells display increased OXPHOS capacity and that DoxR cells have an elevated usage of glutamine for glutathione synthesis.

2. Further work to establish causality for the specific metabolic alterations in resistance to doxorubicin versus epirubicin.

This is indeed an important point, which we have addressed by targeting a distinct and actionable metabolic vulnerability in doxorubicin versus epirubicin resistance.

Doxorubicin-resistant breast cancer cells are reliant on glutamine to synthesize the antioxidant molecule glutathione de novo, and we now show that they are specifically sensitive to inhibition of this pathway. The glutathione synthesis inhibitor buthionine sulfoximine (BSO) significantly inhibited the growth of DoxR cells in vitro, while having little to no effect on Control and EpiR cells (Figure 6C). Furthermore, as described in point #3 below, DoxR tumors were acutely sensitive to BSO while EpiR tumors were unresponsive (Figure 6E, Author response image 1). These new results clearly demonstrate that doxorubicin and epirubicin resistant cells and tumors have different metabolic vulnerabilities and that it is possible to exploit these to limit the growth of therapy resistant tumors.

We have modified the flow of the manuscript to highlight these new data. In order to further address this point, we have elaborated on mechanisms that may underpin these distinct metabolic alterations in the Discussion section (pages 16-17). Briefly, we posit that breast cancer cells treated with doxorubicin may face a greater oxidative challenge than those treated with epirubicin, which aligns with a greater dependence of doxorubicin-resistant cells on de novo glutathione synthesis.

**Author response image 1. sa1fig1:** 

3. Some additional effort to assess the relevance of these resistance mechanisms in vivo. The reviewers felt that the patient correlation included in the study is contrary to the claim that specific metabolic adaptations arise to each drug, and thus some work to show the main claims hold in some in vivo context is needed.

We agree that further in vivo work would strengthen the central claims of the paper. We now present new data fulfilling this requirement.

As mentioned above, our revised manuscript shows that DoxR cells, which are dependent on glutamine for de novo glutathione synthesis, are specifically sensitive to the inhibition of this pathway by buthionine sulfoximine (BSO). in vitro, DoxR cells were more sensitive to BSO than EpiR and Control cells, at all doses tested (Figure 6C). In vivo, daily BSO treatment effectively reduced the growth of DoxR tumors, but had no significant impact on the growth of EpiR tumors over 20 days (Figure 6E, Author response image 1). These results have been added to the manuscript and are represented in Figure 6. The structure of the Results section, as well as the Discussion section, has also been modified to reflect these new results. The significance of this finding is further highlighted in a new schematic summarizing the central claims of the paper (Figure 6H).

Furthermore, we agree with the reviewers that the patient data used were not a perfect model of resistance to anthracyclines, and have removed all pathway analyses performed on this dataset from the manuscript. We also recognize that the concept that both similarities and differences exist in the mechanisms through which DoxR and EpiR cell lines are resistant to their respective anthracycline drug (doxorubicin or epirubicin) was not presented clearly. The manuscript has therefore been significantly revised to address this point.

We now show that doxorubicin- and epirubicin-resistant cells rely on distinct metabolic adaptations, while also exhibiting a considerable level of overlap (over 55%) in their signatures of differentially expressed genes compared to parental Control cells. This overlap is in line with the chemical similarity and mechanism of action of both anthracycline drugs and accordingly, both DoxR and EpiR cells display enrichment in drug clearance pathways and depletion of pathways supporting proliferation compared to Control cells. Given this overlap, both DoxR and EpiR gene expression signatures were found to overlap with the gene expression signature of patient biopsies after treatment with a multi-drug chemotherapy regimen including epirubicin. These results highlight common mechanisms of adaptation to anthracyclines as well as the clinical relevance of the EpiR and DoxR breast cancer cell models.

Despite these similarities, 44% of the gene expression signatures of DoxR and EpiR cells were different, revealing distinct metabolic pathways enriched in either model. These distinct metabolic adaptations are the focus of our story, and are further explored through transcriptomics, metabolomics, and functional genomics analyses. It is therefore accurate to state that there are both similarities and differences in the mechanisms through which DoxR and EpiR cell lines are resistant. Taken together, our experimental evidence demonstrates that (1) in alignment with previously published results, breast cancer cells rely on similar mechanisms such as increased drug efflux, elevated lysosomal activity, and activation of the NFE2L2 pathway to develop resistance to either doxorubicin or epirubicin, and that (2) breast cancer cells can adopt distinct metabolic adaptations to support resistance to either doxorubicin or epirubicin, notably dependence on glutathione metabolism for doxorubicin resistance and mitochondrial OXPHOS capacity for epirubicin resistance. Importantly, we now demonstrate that it is possible to target these different primary metabolic dependencies to limit the growth of therapy resistant cancer, both in vitro and in vivo.

Reviewer #1:

The authors select for doxorubicin and epirubicin resistant MCF7 cells in culture and determine that the global gene expression profile of these cells is similar to tumors from patients exposed to anthracylins. They then characterize the resistant cells, and argue that doxorubicin or epirubicin select for different adaptations, which is surprising given the similarity in these drugs.1. Is the difference between doxorubicin response and epirubicin response simply drug dose? The bulk of the analysis largely compares a cells from a single dose where doxorubicin is cytostatic and epirubicin is cytotoxic. If doses of either drug are titrated to show cytostasis or cytotoxicity is there still a difference in metabolic adaptation of the resistant cells? This is only partially addressed in Figure 7, and is a crucial questions that is important to judge the central claim of the paper.

We acknowledge the concerns of the reviewer and we now provide clarifications regarding the cell line models.

The DoxR and EpiR cell models were first described by the Parissenti group in a 2008 study as outlined in our Methods section (https://dx.doi.org/10.1186%2F1471-2407-8- 318), and have been used in several publications over the years (for example: https://dx.doi.org/10.1158%2F0008-5472.CAN-16-0774, https://dx.doi.org/10.1038%2Fs41598-018-23496-y, https://bmccancer.biomedcentral.com/articles/10.1186/s12885-016-2790-3). These models were selected over hundreds of passages over a twelve-dose process. Doxorubicin-resistant cells (DoxR, originally named MCF-7DOX-2) were selected up to the highest concentration of drug at which the cells could survive (the maximally tolerated dose, 98.1 nM doxorubicin), whereas Epirubicin-resistant cells (EpiR, originally named MCF-7EPI) were still viable at the concentration reached at step twelve (852 nM epirubicin).

As shown by the Parissenti’s group, the initial acquisition of resistance to doxorubicin and epirubicin was linked to reduction in drug accumulation by the cells. However, at higher doses the development of resistance was not linked to drug exclusion by the cells. Indeed, the Figure 2 from Hembruff et al., 2008 clearly shows that the magnitude of resistance was not linked to reduction in drug accumulation after the initial acquisition, as indicated by the plateau.

Furthermore, forcing the accumulation of doxorubicin in DoxR cells using Cyclosporin A (CsA) did not significantly impact the sensitivity of DoxR cells (solid and dotted blue lines in Figure 5A, B in Hembruff et al., 2008). These data clearly highlight the importance of adaptation mechanisms separate from that of the ABC transporters.

These elements considered, and given the fact that DoxR cells are more resistant todoxorubicin (resistance factor, RF=27.8) than epirubicin (RF=4.79), and that EpiR cells are more resistant to epirubicin (RF=815.3) than doxorubicin (RF=203.4), we

hypothesized that the mechanisms of resistance may differ for doxorubicin and epirubicin, despite the chemical similarity of the drugs.

Third and finally, we addressed this difference in the dose by independently deriving cells resistant to 100nM of either doxorubicin or epirubicin. With these cells, we recapitulated the main findings from DoxR and EpiR cells, upholding the conclusions of the paper, i.e. that there are some similar adaptations, as well as unique adaptations, such as increased OXPHOS capacity in the EpiR cells and the elevated usage of glutamine for glutathione synthesis in DoxR cells.

2. There is a large literature arguing that response to chemotherapy in culture is not necessarily indicative of response in patients, and there is a growing literature around cancer metabolism also being different in the cell culture and in tumors, including glutamine metabolism. The bulk of the analysis is based on cell culture, and if some data were available that speak to whether these same differences or dependencies hold in vivo it would greatly improve the manuscript.

We thank the reviewer for this question, we agree that cell culture models can only provide a partial picture of therapeutic response and of the metabolic state. Our study reveals unique metabolic vulnerabilities in DoxR and EpiR cells, which we have further confirmed through the use of a targeted drug both in vitro and in vivo. Indeed, we now show that DoxR cells, which are dependent on glutamine to synthesize the antioxidant molecule glutathione, are specifically sensitive to the inhibition of the γ-glutamylcysteine synthetase enzyme by buthionine sulfoximine (BSO, https://doi.org/10.1016/0006-2952(84)90598-7).

In vitro, DoxR cells were more sensitive to BSO than EpiR and Control cells, at all doses tested (Figure 6C). in vivo, daily BSO treatment effectively reduced the growth of DoxR tumors, but had no significant impact on the growth of EpiR tumors over 20 days (Figure 6E, Author response image 1). The significance of this finding is further highlighted in a new schematic summarizing the central claims of the paper (Figure 6H). These results have been added to the manuscript and are represented in Figure 6.

3. The authors assess labeling of metabolites at short time points (15 min), which can be more useful for assessment of flux through pathways like glycolysis. However, this data would be much more informative if there were multiple time points assessed, as this is typically need to interpret kinetic tracing data.

We agree with the reviewer that multiple time points should be assessed for any tracing experiment, in accordance with the standard practice in the field (A roadmap for interpreting 13C metabolite labeling patterns from cells, https://dx.doi.org/10.1016%2Fj.copbio.2015.02.003). For this reason, all tracing experiments in this study were performed at multiple time points, and complete tracing data are provided in Figure 2—figure supplements 1 and 2, for glucose and glutamine tracing.

4. Seahorse assays were used to suggest ATP production rate, but this approach requires several assumptions that should be more explicitly discussed. I suggest showing the raw data, with or without the calculated rates, and feel this is particularly important for a broad audience who may not appreciate the caveats associated with this analysis.

We agree with the reviewer that these analyses entail several assumptions; this has been made more explicit in the manuscript text (page 9, last paragraph). Raw data for these analyses are OCR and ECAR data, which are presented alongside the ATP production rate results in Figure 3 (Figure 3a-c). OCR and ECAR data for all bioenergetics experiments have now been added to the manuscript, in Figures 4c and S3f (glutamine deprivation experiments) and S5a,b (new resistant models).

5. While the fact that targeting PGC1a overcomes resistance to both doxorubicin and epirubicin is potentially relevant for clinical medicine, from a mechanistic standpoint it does not support their claim that these drugs elicit different metabolic adaptations. Is there an example of a target that separates the contexts as evidence of differential resistance mechanism?

We would like to thank the reviewer for this comment and apologize for any confusion caused by the previous flow of the manuscript. This has been updated to further emphasize the fact that there are indeed similarities between the resistant models, but that this study focused on the differences in order to detect potential vulnerabilities specific to either drug. In this revised manuscript, we further show that only doxorubicin-resistant cells are sensitive to glutathione synthesis inhibition, via the γ-glutamylcysteine inhibitor BSO (buthionine sulfoximine). Compared to both Control and EpiR cells, DoxR cells were acutely sensitive to BSO treatment in vitro (Figure 5f). The growth of DoxR tumors in vivo was significantly decreased by daily injections of BSO, compared to vehicle control, whereas BSO had little to no impact on the growth of EpiR tumors (Figure 5g-i). These results are in accordance with the integrated transcriptomics and metabolomics analyses (Figure 1i), as well as the functional broad shRNA screening (Figure 1j-k) and tracing experiments (Figure 2b-f, Figure S2), showing that glutathione metabolism and glutamine flux are crucial for resistance to doxorubicin, but not to epirubicin.

Our data support the notion that PGC1a is an important player in the resistance of both drugs. PGC-1α is a well-known regulator of mitochondrial biogenesis (Wu, 1999, https://doi.org/10.1016/s0092-8674(00)80611-x), OXPHOS (Mootha, 2003, https://doi.org/10.1038/ng1180), glutamine metabolism (McGuirk, 2013, https://doi.org/10.1186/2049-3002-1-22) and glutathione synthesis (Guo, 2018, https://doi.org/10.1016/j.nbd.2018.02.004). In line, EpiR cells had a significantly greater expression of PGC-1α (Figure 3j), commensurate with their elevated mitochondrial volume and OXPHOS rates, and knockdown of PGC-1α/β also significantly decreased the expression of glutathione synthesis genes particularly in DoxR cells (Figure 3o). Knockdown experiments shown in Figure 3l-n further present evidence that PGC-1α is required for survival of both resistant lines in the presence of their respective drug.

Reviewer #2:

In their manuscript, "Tailored Metabolic Adaptations Confer Resistance to Chemotherapy in Breast Cancer", McGuirk et al. derive breast cancer cells resistant to doxorubicin and epirubicin and investigate whether the metabolic alterations that occur during resistance are dependent on the therapeutic. They suggest that doxorubicin resistant cells rely on glutamine for mitochondrial function and glutathione synthesis, while epirubicin resistant cells increase their mitochondrial capacity and ATP production. They suggest that targeting PGC1a could eradicate cells resistant to both therapeutics, as PGC1a regulates both metabolic adaptions. While the idea that different therapeutics of the same class could cause resistance due to different metabolic mechanisms, this study was limited by a lack of evidence for a causal role for the metabolic alterations identified in drug resistance.1. It was difficult to follow the claims of the manuscript. Many statements are vague and often the figure legends and section headings/text do not match, and are sometimes contradictory in their claims.

We have revised the text of the manuscript and figure legends to ensure proper flow and clarity.

2. The authors generate doxorubicin resistant (DoxR) and epirubicin resistant (EpiR) resistant cell lines via exposure to increasing concentrations of drug. DoxR resistant cells are cultured in a final concentration of 98.1nM drug, while epirubicin resistant cells are cultured in 852nM drug. Treatment of chemonaive cells with these concentrations results in cytostatic (Dox) or cytotoxic (Epi) effects. This approach raises several concerns.a. First, why these concentrations? These differing concentrations are a concern for the major findings of the manuscript.

We acknowledge the concerns of the reviewer and we now provide clarifications regarding the cell line models.

The DoxR and EpiR cell models were first described by the Parissenti group in a 2008 study as outlined in our Methods section (https://dx.doi.org/10.1186%2F1471-2407-8318), and have been used in several publications over the years (for example:

https://dx.doi.org/10.1158%2F0008-5472.CAN-16-0774, https://dx.doi.org/10.1038%2Fs41598-018-23496-y,

https://bmccancer.biomedcentral.com/articles/10.1186/s12885-016-2790-3). These models were selected over hundreds of passages over a twelve-dose process. Doxorubicin-resistant cells (DoxR, originally named MCF-7_DOX-2_) were selected up to the highest concentration of drug at which the cells could survive (the maximally tolerated dose, 98.1 nM doxorubicin), whereas Epirubicin-resistant cells (EpiR, originally named MCF-7_EPI_) were still viable at the concentration reached at step twelve (852 nM epirubicin).

As shown by the Parissenti’s group, the initial acquisition of resistance to doxorubicin and epirubicin was linked to reduction in drug accumulation by the cells. However, at higher doses the development of resistance was not linked to drug exclusion by the cells. Indeed, Figure 2 from Hembruff et al., 2008 clearly shows that the magnitude of resistance was not linked to reduction in drug accumulation after the initial acquisition, as indicated by the plateau.

Furthermore, forcing the accumulation of doxorubicin in DoxR cells using Cyclosporin A (CsA) did not significantly impact the sensitivity of DoxR cells (solid and dotted blue lines in Figure 5A, B in Hembruff et al., 2008). These data clearly highlight the importance of adaptation mechanisms separate from that of the ABC transporters.

These elements considered, and given the fact that DoxR cells are more resistant to doxorubicin (resistance factor, RF=27.8) than epirubicin (RF=4.79), and that EpiR cells are more resistant to epirubicin (RF=815.3) than doxorubicin (RF=203.4), we hypothesized that the mechanisms of resistance may differ for doxorubicin and epirubicin, despite the chemical similarity of the drugs.

Third and finally, we addressed this difference in the dose by independently deriving cells resistant to 100nM of either doxorubicin or epirubicin. With these cells, we recapitulated the main findings from DoxR and EpiR cells, upholding the conclusions of the paper, i.e. that there are some similar adaptations, as well as unique adaptations, such as increased OXPHOS capacity in the EpiR cells and the elevated usage of glutamine for glutathione synthesis in DoxR cells.

b. Second, anthracyclins have been shown to damage mitochondrial DNA, and the use of differing concentrations may result in different amounts of damage. Related to this point does compensatory mitochondrial biogenesis explain why PGC1a expression is increased, particularly in the EpiR cells, which are treated with significantly more anthracycline? What happens to mitochondrial content in these resistant cell lines? Mitochondrially encoded respiratory components? How does PGC1a/b knockdown affect mitochondrial content?

We thank the reviewer for this very interesting question. As stated in (a), the intracellular concentration of drug is known to be quite lower than extracellular concentrations, and given that these are established models, we do not know whether mitochondrial damage may have occurred earlier in the temporal acquisition of resistance.

Through immunofluorescence experiments, we have confirmed that EpiR cells indeed have elevated mitochondrial content compared to Control cells, commensurate with their increased reliance on OXPHOS and their elevated levels of PGC-1α expression (see Figure 3i). The mitochondrial content of DoxR cells is not significantly different from that of Control cells.

We also agree with the reviewer that the mitochondrial content would be expected to decrease upon PGC1a knockdown. However, we did not pursue the quantification of mitochondrial content in PGC1a KD cells considering that this experiment would not further enhance the narrative of the revised paper which focuses more specifically on targeting the differential metabolic vulnerability of doxorubicin-resistant cells through inhibition of glutathione synthesis.

c. Third, the final figure, in which new resistant cell lines are derived and grown in the same amount of Dox or Epi, the authors show now that while DoxR cells still increase GSH synthesis from glucose, similar increases in mitochondrial ATP and bioenergetic capacity are observed between DoxR and EpiR cells. While the effects are modestly more pronounced in EpiR cells, this finding is contrary to the thesis that Dox and Epi resistance induce different metabolic alterations as it relates to the mitochondria. These cell lines should be used throughout the study to eliminate any confounding effects of differing drug concentrations.

Please refer to our extensive description of the DoxR and EpiR cells and the different drug concentrations in response to point #2a.

It is expected that there will be some variation in the derivation of an independent model of therapeutic resistance. However, the most important point is that this new model confirms the main findings of our paper: PGC-1α is implicated in anthracycline resistance, elevated glutamine-derived de novo glutathione is a key feature of doxorubicin-resistant models, and epirubicin-resistant models have markedly elevated mitochondrial bioenergetic capacity. The reliance of doxorubicin-resistant cells on glutathione synthesis is the particular highlight of the revised manuscript, which was further confirmed in vitro and in vivo through targeted therapy with γ-glutamylcysteine inhibitor BSO (buthionine sulfoximine). Compared to both Control and EpiR cells, DoxR cells were acutely sensitive to BSO treatment in vitro (Figure 5f). The growth of DoxR tumors in vivo was significantly decreased by daily injections of BSO, compared to vehicle control, whereas BSO had little to no impact on the growth of EpiR tumors (Figure 5g-i).

One particular difference between the experimental models of resistance is that PGC-1α expression is more elevated in the MCF-7 cells resistant to 100nM doxorubicin (D100) than it is in DoxR compared to their respective parental lines, which likely explains the elevated basal OCR in D100 cells compared with DoxR cells (Figure S4b versus Figure 3j, Figure S5a versus Figure 3b). Nevertheless, Epi100 cells still display elevated basal OCR and much greater mitochondrial bioenergetic capacity (Max J_ATPox_, Figure S4d) compared to D100 cells analogous to the EpiR cells compared with the DoxR cells (Figure S4c-f versus Figure 3b,f,g).

The figures displaying the new resistant cell lines was moved into supplementary, given that its purpose is solely to confirm the key findings in an independent model.

3. Related to the point above, it is unclear how PGC1a is promoting resistance to doxorubicin and epirubicin. In Figure 6, PGC1a associates with the antioxidant response genes only in EpiR cells, but PGC1a/b knockdown has a greater effect on gene expression in the DoxR cells.

The figure described by the reviewer was only intended to show that PGC-1α can bind to the promoters of multiple genes (ChIP-qPCR) described in the paper, particularly the glutathione metabolism pathway genes. Given the low expression of PGC-1α in Control and DoxR cells, its enrichment at the promoters of genes in these cell lines is low. The loading was clear in EpiR cells, given the relatively high (11-fold) expression of PGC-1α in this model compared to Control and DoxR cells. As pointed out by the reviewer, the magnitude of binding does not necessarily scale with gene expression. These panels have been moved into supplemental Figure S3b,c.

PGC-1α is well known to regulate numerous metabolic programs, in a context-dependent manner. For example, PGC-1α regulates mitochondrial biogenesis (Wu, 1999, https://doi.org/10.1016/s0092-8674(00)80611-x), OXPHOS (Mootha, 2003, https://doi.org/10.1038/ng1180), glutamine metabolism (McGuirk, 2013, https://doi.org/10.1186/2049-3002-1-22) and glutathione synthesis (Guo, 2018, https://doi.org/10.1016/j.nbd.2018.02.004). Indeed, EpiR cells had a significantly greater expression of PGC-1α (Figure 3j), commensurate with their elevated mitochondrial volume and OXPHOS rates. Knockdown of PGC-1α/β also significantly decreased the expression of glutathione synthesis genes particularly in DoxR cells (Figure 3o). Data shown in Figure 3l-n further present evidence that PGC-1α is required for survival of both resistant lines in the presence of their respective drug, despite different specific metabolic adaptations.

What is the effect of PGC1a knockdown alone?

Based on our experience working with the PGC-1 family of transcriptional co-activators, we typically knockdown the two main isoforms because there are usually compensation mechanisms between the two (https://doi.org/10.1111/febs.13175) due to great overlap in their functions (https://doi.org/10.1016/j.cmet.2005.05.004).

In addition, Figure 3i shows that EpiR cells have little to no elevation of antioxidant gene expression, despite significantly increased PGC1a binding to their promoters. How are these findings reconciled?

The expression of antioxidant genes was, in fact, significantly elevated when compared to Control cells for both DoxR and EpiR, albeit more modestly in the latter (1.5 – 3 fold, see Figure 2J). Specifically, NFE2L2, NQO1, GPX3, PRDX5, and CAT all have elevated expression in EpiR compared to Control. As pointed out by this reviewer in a previous point, the magnitude of binding does not necessarily scale with gene expression. As noted above, the figure detailing promoter binding has been moved to Figure S3b-c.

Further, siPGC1a/b affects the growth of DoxR and EpiR cells equally. Is this in the presence of drug? It is unclear from the figure legend.

Indeed, this experiment was performed while cells were in presence of each respective drug (EpiR with epirubicin, DoxR with doxorubicin). We state in the Methods section that these cells are always kept in presence of the drug throughout all experiments presented in the paper; for clarity we reiterated the presence of anthracyclines in the figure legend for Figure 3m as well.

Finally, is PGC1a overexpression sufficient to promote resistance of drug naïve cells? This experiment is needed to show causation.

While this experiment is certainly interesting, we focused the revised version of the manuscript on the different metabolic adaptations of EpiR and DoxR cells with a notable emphasis on the glutathione metabolism dependence of DoxR cells.

4. The authors suggest that cells with engineered resistance to Dox and Epi have similar gene expression patterns to residual tumors of breast cancer patients following anthracycline-based chemotherapy, but there are several problems with this analysis/these claims. First, it is not clear that these tumors are resistant to anthracyclines. From the description, patients received 4 cycles of anthracycline-based therapy, followed by four cycles of docetaxel. Resultant tumors may be responders, resistant to anthracyclines, resistant to docetaxel, or both. As a consequence, it is difficult to draw meaningful conclusions from the comparison. Second, the authors compare their cell line gene expression to these tumors, but tumors were treated only with epirubicin. Since the authors argue that resistance to Dox vs. Epi is via discrete mechanisms, the argument for similarity between DoxR cells and Epi "resistant" tumors is contrary to the main claims of the manuscript.

We agree with the reviewer that the patient data used were not a perfect model of resistance to anthracyclines, and have removed all pathway analyses performed on this dataset from the manuscript. We also recognize that the concept that both similarities and differences exist in the mechanisms through which DoxR and EpiR cell lines are resistant to their respective anthracycline drug (doxorubicin or epirubicin) was not presented clearly. The manuscript has therefore been significantly revised to address this point.

We now show that doxorubicin- and epirubicin-resistant cells rely on distinct metabolic adaptations, while also exhibiting a considerable level of overlap (over 55%, as shown in Figure 1d) in their signatures of differentially expressed genes compared to parental Control cells. This overlap is in line with the chemical similarity and mechanism of action of both anthracycline drugs, and accordingly both DoxR and EpiR cells display enrichment in drug clearance pathways and depletion of pathways supporting proliferation compared to Control cells. Given this overlap, both DoxR and EpiR gene expression signatures were found to overlap with the gene expression signature of patient biopsies after treatment with a multi-drug chemotherapy regimen including epirubicin. These results highlight common mechanisms of adaptation to anthracyclines as well as the clinical relevance of the EpiR and DoxR breast cancer cell models.

Despite these similarities, 44% of the gene expression signatures of DoxR and EpiR cells were different, revealing distinct metabolic pathways enriched in either model. These distinct metabolic adaptations are the focus of our story, and are further explored through transcriptomics, metabolomics, and functional genomics analyses. It is therefore accurate to state that there are both similarities and differences in the mechanisms through which DoxR and EpiR cell lines are resistant. Taken together, our experimental evidence demonstrates that (1) in alignment with previously published results, breast cancer cells rely on similar mechanisms such as increased drug efflux, elevated lysosomal activity, and activation of the NFE2L2 pathway to develop resistance to either doxorubicin or epirubicin, and that (2) breast cancer cells can adopt distinct metabolic adaptations to support resistance to either doxorubicin or epirubicin, notably dependence on glutathione metabolism for doxorubicin resistance and mitochondrial OXPHOS capacity for epirubicin resistance. Importantly, we now demonstrate that it is possible to target these different primary metabolic dependencies to limit the growth of therapy resistant cancer, both in vitro and in vivo.

Indeed, the claim that "several distinct metabolic pathways were altered by doxorubicin or epirubicin resistance (Figure 1f,h)" is problematic due to these concerns since (1) these pathways are based on overlap with gene expression from tumors from patients that may or may not be resistant and (2) these tumors were treated only with epirubicin. Therefore, the basis for distinct metabolic pathways is unsound.

As stated above, in agreement with the reviewer, we have removed the pathway analysis from the manuscript.

5. The increase in GCLC, ME1, NQO1 etc in DoxR cells is consistent with NRF2 activation as proposed, as is the increase in GSH M+5, which suggests an increase in GSH synthesis. However, the decrease in the GSH/GSSG ratio and NADPH, and increase in NADP+ is not consistent, nor is it consistent with the protection against ROS shown in Figures 3j and 3k. How is this reconciled?

We thank the reviewer for this comment. In order to validate our results, we quantified GSH and GSSG using an alternative method that is particularly suited for metabolites that are easily oxidized like GSH. These new analyses show that the GSH:GSSG ratio in DoxR is higher than Control and EpiR cells. This novel extraction method revealed that there was oxidation of GSH in our original datasets, explaining the considerable variability in the absolute amounts of GSH and the GSH/GSSG ratios. Stable isotope tracing experiments were unaffected by this oxidation, and there was little variability in the fractional enrichment measured across all experiments. The new results have been incorporated in Figure 2H. These data align with the decrease in NADPH:NADP ratio (Figure S3g), likely due to increased Glutathione Reductase activity, and high GSH levels providing a protection against ROS.

What is the total pool size of GSH + GSSG?

The total pool of glutathione is bigger in drug-resistant cells and as expected, DoxR cells have the biggest pool (see Figure 2G).

Related to this, what is the "control" for figures 3J and 3k? Parental cells?

We apologize for the confusion. The parental cells are called Control, and we ensure this word was capitalized in the legend to indicate that this refers to the cell line. For clarity, we have changed the reference to “Control” in the legend to “Control cells”.

6. The data shown in Figure 4 is not supportive of an increased dependence of DoxR cells on glutamine, as claimed in the figure legend. Glutamine is used for many purposes in cells and neither DoxR nor EpiR cells grow without glutamine in Figure 4A. Further, the idea that glutamine is driving TCA cycle is not matched by the tracing data from Figure 3. There is no increased entry of glucose or glutamine into the TCA cycle in the DoxR cells (e.g. citrate m+2 from glucose, citrate m+4 from glutamine).

We have corrected the figure legend to clarify that DoxR cells have increased dependence on glutamine for mitochondrial ATP production. Indeed, while it is true that the proliferation of both DoxR and EpiR cells is arrested upon glutamine withdrawal (Figure 4a), only DoxR cells face a severe bioenergetic challenge as a result. Their bioenergetic capacity is drastically reduced (Figure 4g-j), and their basal mitochondrial ATP production is significantly decreased (Figure 4f).

Figure 3 shows that glutamine is utilized at similar levels in the TCA in DoxR, EpiR, and Control cells, There is indeed little evidence of increased entry of glutamine into the TCA in DoxR compared to Control cells, albeit a slight increase in glutamine flux to succinate in the mitochondria shown in time-course tracing diagrams in Figure S3c. Figure 4, on the other hand, includes a functional assay which demonstrates that while glutamine withdrawal impacts the growth of all cell types, it specifically and significantly impairs mitochondrial ATP production in DoxR cells – a bioenergetic process that is dependent on the TCA cycle.

Regarding the entry of glucose into the TCA cycle, glucose carbons are indeed diminished at m+2 citrate in DoxR cells compared to Control cells. However, there is increased m+3 labeling in citrate, possibly due to pyruvate carboxylase activity. In addition, m+3 malate is more elevated than m+2 malate is decreased, showing a small but net increase in glucose flux into the TCA cycle in DoxR cells compared to Control cells (see Figures 3c-d and S3a).

Related to this, the inability of DoxR cells to compensate for ATP production upon glutamine withdrawal suggests a defect in glycolytic regulation or ATP sensing, rather than glutamine metabolism. More clarity is needed for (1) what metabolic alterations occur, (2) how they are regulated and (3) how they promote anthracycline resistance.

We thank the reviewer for the suggestions related to glycolytic regulation and ATP sensing, and have now elaborated on these results in the Results (page 12, paragraph 2) and Discussion (page 16, paragraph 1) sections of the manuscript. We did not pursue further experiments on this aspect as the revised version of the manuscript is now centred on the dependence of DoxR cells on glutathione metabolism considering the editorial and reviewers’ recommendations.

Reviewer #3:

In "Tailored Metabolic Adaptations Confer Resistance to Chemotherapy in Breast Cancer", McGuirk et al. use MCF-7 cells in culture to study mechanisms of resistance to doxorubicin and epirubicin. Overall, I found their approach to be intriguing. It seems like a compelling approach to study drug resistance in cancer. They subjected cells to many passages in the presence of drug, which resulted in a population of resistant cells. They could then compare these resulting resistant cells to the original drug-susceptible cells. In doing so, the authors claim to have found different metabolic adaptations to each drug. Given the differences in adaptation, they suggest that it might be better to target resistant tumors with drugs that hit global regulators of metabolism, like PGC-1a.I have two major questions.1. How do the authors know that the genes and metabolic adaptations assumed in the resistant cells are essential to drug resistance?

We show several lines of evidence that support the role of metabolic genes in sustaining drug resistance. In our broad shRNA screen (Figure 1j), in which depleted shRNA barcodes indicate gene targets whose knockdown impairs growth and/or viability, glutathione metabolism genes were found to be particular targets in DoxR cells (Figure 1k). It was further determined through functional assays that glutamine carbons are highly utilized for glutathione synthesis (Figure 2f) and are necessary to sustain mitochondrial ATP production specifically in DoxR cells (Figure 4c,d,f,g,i). Finally, we have performed new experiments and found that only doxorubicin-resistant cells are sensitive to glutathione synthesis inhibition, via the γ-glutamylcysteine inhibitor BSO (buthionine sulfoximine). Compared to both Control and EpiR cells, DoxR cells were acutely sensitive to BSO treatment in vitro (Figure 5f). The growth of DoxR tumors in vivo was significantly decreased by daily injections of BSO, compared to vehicle control, whereas BSO had little to no impact on the growth of EpiR tumors (Figure 5g-i). The significance of this finding is further highlighted in a new schematic summarizing the central claims of the paper (Figure 6H).

On the other hand, the broad shRNA screen found that oxidative phosphorylation genes were key knockdown targets impairing the growth and/or viability of EpiR cells. We confirmed through functional assays that EpiR cells have elevated mitochondrial respiration rates compared to both Control and DoxR cells (Figure 3a,b), and that they have a markedly increased mitochondrial volume (Figure 3i) and OXPHOS capacity (Figure 3f-h).

We proposed that PGC-1α contribute to these distinct adaptations, given that it is known to regulate mitochondrial biogenesis (Wu, 1999, https://doi.org/10.1016/s00928674(00)80611-x), OXPHOS (Mootha, 2003, https://doi.org/10.1038/ng1180), and glutathione synthesis (Guo, 2018, https://doi.org/10.1016/j.nbd.2018.02.004). PGC-1α is also well known to have context-dependent roles in different tissues, cancer types, and in response to different treatments / stimuli (https://dx.doi.org/10.3389%2Ffonc.2018.00075). Indeed, EpiR cells had a significantly greater expression of PGC-1α (Figure 3j), commensurate with their elevated mitochondrial volume and OXPHOS rates, and knockdown of PGC-1α/β also significantly decreased the expression of glutathione synthesis genes particularly in DoxR cells (Figure 3o). Knockdown experiments shown in Figure 3l-n further present evidence that PGC-1α is required for survival of both resistant lines in the presence of their respective drug, despite different specific metabolic adaptations downstream.

Might it be the case that their original control population of cells was heterogeneous and exposure of cells to drug over long periods of time simply selected those subclones that were resistant? In such a model, while one gene/metabolic phenotype may be essential to developing resistance (perhaps ABC transporters), the other genes/metabolic phenotypes could be random. Indeed, based on the limited amount of data shown from two experiments, it seems like the metabolic phenotypes of cells exposed to dox for long periods of time are different in separate experiments. For example, the isotope labeling shown in Figure 3 looks very different from the labeling shown in Figure 7. A recent study by Speirs and Price et al. ( doi: 10.18632/oncotarget.26533 ) did a similar experiment without drugs. After selecting subclones of cells from a culture, they found significant differences in metabolic phenotypes. It would be most interesting to compare the genetic and metabolic differences of resistance cells that were selected independently by drugs in separate experiments. This could help better resolve whether the differences observed are essential.

We thank the reviewer for this important comment; this is indeed why we developed the secondary model cell lines used in Figure 7 (now, moved to Figures S4 and S5). The models that we used for most of the study (DoxR, and EpiR) were selected by the Parissenti group. These models were selected over hundreds of passages over a twelvedose process. Doxorubicin-resistant cells (DoxR, originally named MCF-7_DOX-2_) were selected up to the highest concentration of drug at which the cells could survive (the maximally tolerated dose, 98.1 nM doxorubicin), whereas Epirubicin-resistant cells (EpiR, originally named MCF-7_EPI_) were still viable at the concentration reached at step twelve (852 nM epirubicin). Control cells (originally named MCF-7_CC_) were developed in parallel, through passaging with a constant dose of DMSO (https://dx.doi.org/10.1186%2F1471-2407-8-318). A subclonal selection likely occurred over time through long-term passaging of these cells, either through adaptation of certain cells to the drug or due to inherent resistance in subclones. For this reason, we expected that there may be some differences between these cells and those we derived. Indeed, when comparing differential gene expression profiles between the two models of doxorubicin resistance, and between the two models of epirubicin resistance, there is a large 66% overlap in both cases (see Author response image 2). Nevertheless, one-third of differentially expressed genes are distinct between D100 and DoxR, and between E100 and EpiR (see Author response image 2). Importantly and despite these clonal differences, these new cells confirmed that glutamine metabolism through glutathione is specifically important in doxorubicin-resistant cells, and that epirubicin-resistant cells have especially elevated OXPHOS bioenergetic capacity. The metabolic differences in doxorubicin and epirubicin resistant cells is key for their resistance and this is well illustrated by the fact that doxorubicin resistant cells are more sensitive than epirubicin resistant cells to the drug BSO, that interferes with glutathione synthesis (see graphs in Figure 6).

**Author response image 2. sa1fig2:** Venn diagrams of differential gene expression, comparing D100 vs Ctl and DoxR vs Control cells (left) and comparing E100 vs Ctl and EpiR vs Control cells (right).

2. The major premise of this manuscript is that the authors have selected cells resistant to dox and epi. Data shown in Figure 1B suggest that this is the case over 6 days. Based on the data shown in Figure 4A, however, I worry that the Epi-R cells are not resistant to Epi over long periods of time. For example, at day 11, the Epi-R live cell count looks more like the control cells in Figure 1 than DoxR. Thus, if the cells are not both equally resistant, maybe the metabolic adaptations are not really "distinct" but rather just reflective of different degrees of resistance.

We acknowledge the concerns of the reviewer and we now provide clarifications

regarding the cell line models.

The DoxR and EpiR cell models were first described by the Parissenti group in a 2008 study as outlined in our Methods section (https://dx.doi.org/10.1186%2F1471-2407-8-318), and have been used in several publications over the years (for example:

https://dx.doi.org/10.1158%2F0008-5472.CAN-16-0774, https://dx.doi.org/10.1038%2Fs41598-018-23496-y, https://bmccancer.biomedcentral.com/articles/10.1186/s12885-016-2790-3). These models were selected over hundreds of passages over a twelve-dose process. Doxorubicin-resistant cells (DoxR, originally named MCF-7DOX-2) were selected up to the highest concentration of drug at which the cells could survive (the maximally tolerated dose, 98.1 nM doxorubicin), whereas Epirubicin-resistant cells (EpiR, originally named MCF-7EPI) were still viable at the concentration reached at step twelve (852 nM epirubicin).

As shown by the Parissenti’s group, the initial acquisition of resistance to doxorubicin and epirubicin was linked to reduction in drug accumulation by the cells. However, at higher doses the development of resistance was not linked to drug exclusion by the cells. Indeed, Figure 2 from Hembruff et al., 2008 clearly shows that the magnitude of resistance was not linked to reduction in drug accumulation after the initial acquisition, as indicated by the plateau.

Furthermore, forcing the accumulation of doxorubicin in DoxR cells using Cyclosporin A (CsA) did not significantly impact the sensitivity of DoxR cells (solid and dotted blue lines in Figure 5A, B in Hembruff et al., 2008). These data clearly highlight the importance of adaptation mechanisms separate from that of the ABC transporters.

These elements considered, and given the fact that DoxR cells are more resistant to doxorubicin (resistance factor, RF=27.8) than epirubicin (RF=4.79), and that EpiR cells are more resistant to epirubicin (RF=815.3) than doxorubicin (RF=203.4), we hypothesized that the mechanisms of resistance may differ for doxorubicin and epirubicin, despite the chemical similarity of the drugs.

Third and finally, we addressed this difference in the dose by independently deriving cells resistant to 100nM of either doxorubicin or epirubicin. With these cells, we recapitulated the main findings from DoxR and EpiR cells, upholding the conclusions of the paper, i.e. that there are some similar adaptations, as well as unique adaptations, such as increased OXPHOS capacity in the EpiR cells and the elevated usage of glutamine for glutathione synthesis in DoxR cells.

To further address the reviewer’s concern and confirm that these cells are indeed stably resistant, we performed a drug holiday experiment. After 7 weeks of proliferation without drug, both DoxR and EpiR cells fully retained their level of resistance when re-exposed to 98.1 nM doxorubicin and 852 nM epirubicin, respectively – see Figure 1—figure supplement 1A in the resubmitted manuscript.

Other points:1. Can the authors do any experiments to explore (or possibly speculate in the discussion) why the two drugs would elicit different resistance mechanisms?

We thank the reviewer for this point, and have elaborated on this further in the Discussion section of the manuscript (pages 16-17). Briefly, we describe that structural differences between these drugs may play a role, as they may lead to distinct on- and off-target effects including differential rates of drug-induced ROS production. Cardiotoxicity, a common side effect of anthracyclines, is linked to oxidative stress and epirubicin has been shown to induce less cardiotoxic effects than doxorubicin, even if both drugs display equivalent response rate to treat breast cancer (https://doi.org/10.1159/000500204). Breast cancer cells treated with doxorubicin may therefore face a greater oxidative challenge than those treated with epirubicin, which aligns with a greater dependence of doxorubicin-resistant cells on de novo glutathione synthesis.

2. The increase of pyruvate carboxylase in EpiR cells is interesting. What is the phenotypic value of such a metabolic alteration in resistance?

As shown in Figure 2A, expression of pyruvate carboxylase (PC) is increased in DoxR compared to Control (not EpiR, as the reviewer suggests). We suggest that this may contribute to resistance by serving as an alternate pathway for refueling the citric acid cycle, perhaps allowing excess glutamine / glutamate to be used instead for GSH synthesis.

3. The data do not seem to support this statement: "DoxR cells were unable to increase glycolytic ATP production to compensate for diminished oxidative ATP production upon glutamine withdrawal." The Seahorse plots in Figure 4 actually show that glycolytic ATP production increases in drug resistance.

Figure 4f shows that glycolytic ATP production (JATPglyc; panel e) remains unchanged after glutamine withdrawal in DoxR, whereas mitochondrial ATP production (JATPox; panel d) is significantly decreased. These results support that DoxR cells are not able to increase their glycolytic ATP production, resulting to a net reduction in total ATP

production (panel f).

4. It would be helpful to see all of the isotopologues, instead of just a select number. For example, M+3 in malate from 13C-glucose doesn't necessarily mean pyruvate carboxylase activity. It could also be indicative of two rounds of TCA cycle (the first with labeled acetyl-CoA and the second with unlabeled). It would be easier to assess these kinds of possibilities if the authors presented full isotopologue plots.

We agree with the reviewer. In the main figures we show only a few isotopologues for simplicity. We have now provided these data in Figures S2b and S2d (corresponding to tracing experiments in Figure 2) as well as in Figure S5c (corresponding to tracing experiments in Figure S4g).

5. Many of the interesting data shown are not explored (or discussed). For example, in Figure 5C-D, the DoxR cells have uncoupled mitochondria. Why is this? Is this the source of ROS? It might be best to remove data that are not discussed, as it is distracting from the overall message.

We agree with the reviewer that the uncoupled respiration in resistant cells is interesting and we have elaborated on this further in the Discussion section of the manuscript (pages 16-17). We describe that doxorubicin are associated with higher induction of ROS than epirubicin (https://doi.org/10.1074/jbc.m508343200) and that, accordingly, epirubicin has been shown to induce less cardiotoxic effects than doxorubicin (https://doi.org/10.1159/000500204). Breast cancer cells treated with doxorubicin may therefore face a greater oxidative challenge than those treated with epirubicin, which aligns with a greater dependence of doxorubicin-resistant cells on de novo glutathione synthesis. Accordingly, doxorubicin-resistant cells also displayed much greater engagement of oxidative stress response than epirubicin-resistant cells. Comparatively, EpiR cells (not DoxR, as the reviewer suggests) displayed an elevated level of uncoupled respiration, which may represent an alternate approach to minimizing ROS production in this model; uncoupled respiration can be induced by ROS, and can play a role in further minimizing ROS in the mitochondria (https://doi.org/10.1038/415096a). To further outline the level of uncoupled respiration in these cell lines, we provide Author response image 3 displaying the percentage of total oxygen consumption rate (OCR) attributed to uncoupled respiration.

**Author response image 3. sa1fig3:** 

[Editors’ note: what follows is the authors’ response to the second round of review.]

Essential revisions:1. Concern remains regarding how relevant the metabolic differences between these two cell populations are to resistance in patients given that different drug concentrations were used to derive the EpiR and DoxR cells and that the new in vivo data did not use a de novo resistance model, but rather the same cell lines used previously. The use of independently derived cell lines should be used to confirm more of the central findings of the manuscript. In particular, please address whether the different metabolic adaptations are related to the dose or the drug itself.

We would like to thank the reviewers for this constructive comment. Acknowledging these concerns, we have provided further confirmation of our main findings in our independently derived cell lines. In addition to previously corroborated findings that epirubicin-resistant cells display increased OXPHOS capacity and that doxorubicin-resistant cells have an elevated usage of glutamine for glutathione synthesis, these data are now described more prominently in the Results section of the manuscript and in the main figures (Figures 5 and 6).

Specifically, we have recapitulated in vitro that independently derived doxorubicin-resistant cells (D100) are also specifically vulnerable to inhibition of the glutathione synthesis pathway by buthionine sulfoximine (BSO), as they display a significantly greater proliferation inhibition to BSO when compared to epirubicin-resistant cells (Figure 6c,d). We also confirmed that independently derived doxorubicin-resistant cells display a lower ROS signal than epirubicin-resistant cells, both at baseline and after H_2_O_2_ treatment (Figure 5d,e). Finally, following point #3 below and in line with elevated OXPHOS activity in both epirubicin-resistant cell models (EpiR and independently derived E100) compared to drug-sensitive and doxorubicin-resistant cells, we now show that both epirubicin-resistant cell models are specifically sensitive to inhibition of OXPHOS using the biguanide phenformin (Figure 6a,b).

Thereby, we show in two independently derived models not only that distinct metabolic adaptations support resistance to doxorubicin or epirubicin, but also that these adaptations can be specifically targeted using metabolic interventions. Given that these independently derived cell lines (D100 and E100 cells) were selected to a common end-point dose of 100nM of doxorubicin or epirubicin, these data further demonstrate that these metabolic adaptations are specific to the drug, not the dose. New panels (d and e) in Figure 5.

2. The tumor growth differences are not impressive when considering that the DoxR cells grow much more slowly than the EpiR cells. The difference in size even for the DoxR tumors does not seem that great, and is presented as fold change. Please at least acknowledge this, consider presenting the data in a more fair way with tumor size data graphed as actual measured size with the DoxR and EpiR curves on the same plot or plots with the same scales.

We acknowledge that our original data presentation did not show the DoxR and EpiR tumors on the same scale, and now show this data as requested by the reviewers with the actual measured sizes of DoxR and EpiR tumors shown on the same plot (Figure 6f). Importantly, while all EpiR tumors are larger than DoxR tumors, our data clearly show that both types of tumors doubled in size over 20 days when treated with vehicle (from 100mm^3^ to 200mm^3^ for DoxR tumors, and from 325mm^3^ to 650mm^3^ for EpiR tumors, Figures 6g). Growth of DoxR tumors was significantly impaired by BSO treatment, as tumor size increased by only 30% over 20 days, whereas there was little impact of BSO on EpiR tumor growth (Figure 6f-g). Our conclusions remain the same, as daily BSO treatment effectively reduced the growth of DoxR tumors, but had no significant impact on the growth of EpiR tumors over 20 days.

3. Please test (in vitro) whether the oxidative stress response in control cells treated with either doxorubicin or epirubicin is different as a potential mechanism for how treatment with these drugs differs in a way that might impact metabolism. Please also considering examining whether Epic cells are more sensitive to OXPHOS inhibitors to consider whether PGC1a promoting the OXPHOS phenotype in epi resistant cells speaks to the divergent metabolism.

We thank the reviewers for these insightful suggestions and agree that these data would greatly complement our findings. Accordingly, we have performed new experiments showing that, in line with their elevated bioenergetic capacity and mitochondrial ATP production rates, both epirubicin-resistant cell models (EpiR and E100) are significantly more sensitive to inhibition of OXPHOS using the biguanide phenformin than doxorubicin-resistant cells (DoxR and D100 cells, Figures 6a-b).

We also performed CM-H_2_DCFDA experiments to test whether different anthracyclines induce different levels of oxidative stress in drug-sensitive Control cells. While there was a significant induction of oxidative stress when these cells were acutely challenged with either drug, we did not see any significant differences between doxorubicin and epirubicin treated cells. The lack of difference in terms of oxidative stress upon acute exposure to doxorubicin and epirubicin is perhaps not surprising in light of the fact that both resistant models used in our study were selected through a sustained and stepwise increase in drug dose over the course of 8-12 months. Indeed, while oxidative stress upon acute exposure (2-3 days) appears similar with both anthracycline drugs, it likely plays a more critical role under chronic treatment conditions and in the development of resistance to doxorubicin compared to epirubicin as demonstrated by our results using transcriptomics and metabolomics, and validated independently by performing a targeted shRNA screen. The Discussion section of the manuscript has been modified to address these points.

4. Please address the issues with the resistance mechanisms in vivo based on the following comment from Reviewer #3. The in vivo BSO experiment is supportive of the cell culture experiment but there are multiple issues with this experiment. First, no anthracycline is included in this experiment making its relevance to cancer therapy questionable. Second, resistant lines are used from the start, rather than assaying the role of GSH synthesis in de novo resistance that arises in vivo, which would have been more relevant to the patient situation.

The authors thank the reviewer for this comment.

The in vivo BSO experiments conducted during the first round of revisions, to address a specific comment made by the reviewers and editors, were designed to directly address whether doxorubicin-resistant tumors are specifically vulnerable to inhibition of the glutathione synthesis pathway using BSO. The experimental design we adopted enabled us to control for several potentially confounding variables, thereby increasing confidence in the results. First, we wanted to avoid the use of anthracycline drugs in mice given that the cell lines used are resistant to very high doses of anthracyclines, which would be highly toxic to mice. Hence, prior to in vivo experiments, we grew the resistant cell lines in the absence of anthracyclines in vitro, as would be the situation during the xenograft experiments in vivo, and confirmed that the cell models used (DoxR and EpiR) retain their resistance to their respective drug for prolonged periods of time, as shown in Figure S1a. Our experimental design also controlled for

variability between mice, by growing DoxR and EpiR cells in the same mouse. Furthermore, the results of this experiment highlight a potentially clinically relevant finding. Given that anthracycline treatments are linked to significant side effects in patients such as irreversible cardiotoxicity which limits their use to a restrictive cumulative total lifetime dose, there is important clinical relevance in reducing tumor growth in anthracycline-resistant patients through a secondary treatment option without administrating additional anthracycline chemotherapy.

We agree with the reviewer that assaying the role of GSH synthesis in de novo resistance that arises in vivo would provide interesting findings. However, this proposed experiment represents a very significant endeavor which the authors consider to be outside the scope of the present study.